# A single-cell transcriptomic atlas reveals resident dendritic-like cells in the zebrafish brain parenchyma

Mireia Rovira[1,2†], Giuliano Ferrero[1,2†], Magali Miserocchi[1,2], Alice Montanari[1,2], Ruben Lattuca[1,2], Valerie Wittamer[1,2*]

[1]Institut de Recherche Interdisciplinaire en Biologie Humaine et Moléculaire (IRIBHM) - Jacques E. Dumont, Brussels, Belgium; [2]ULB Institute of Neuroscience (UNI), Université Libre de Bruxelles (ULB), Brussels, Belgium

## eLife Assessment

This **important** work represents an advance in our understanding of resident myeloid cells in the zebrafish brain, particularly as it provides a molecular definition of dendritic cell subtypes associated with their localization. Combined evidence from single cell transcriptomics and histology is **compelling**. The associated atlas will be used as a resource by the zebrafish community and beyond.

**\*For correspondence:**
valerie.wittamer@ulb.be

[†]These authors contributed equally to this work

**Competing interest:** The authors declare that no competing interests exist.

**Abstract** Recent studies have highlighted the heterogeneity of the immune cell compartment within the steady-state murine and human CNS. However, it is not known whether this diversity is conserved among non-mammalian vertebrates, especially in the zebrafish, a model system with increasing translational value. Here, we reveal the complexity of the immune landscape of the adult zebrafish brain. Using single-cell transcriptomics, we characterized these different immune cell subpopulations, including cell types that have not been or have only been partially characterized in zebrafish so far. By histology, we found that, despite microglia being the main immune cell type in the parenchyma, the zebrafish brain is also populated by a distinct myeloid population that shares a gene signature with mammalian dendritic cells (DC). Notably, zebrafish DC-like cells rely on *batf3*, a gene essential for the development of conventional DC1 in the mouse. Using specific fluorescent reporter lines that allowed us to reliably discriminate DC-like cells from microglia, we quantified brain myeloid cell defects in commonly used *irf8*[-/-], *csf1ra*[-/-], and *csf1rb*[-/-] mutant fish, revealing previously unappreciated distinct microglia and DC-like phenotypes. Overall, our results suggest a conserved heterogeneity of brain immune cells across vertebrate evolution and also highlights zebrafish-specific brain immunity characteristics.

## Introduction

Over the last years, several landmark studies leveraging high-dimensional techniques have contributed to uncovering the cellular complexity of the human and murine central nervous system (CNS) immune landscapes (*Mrdjen et al., 2018*; *Hammond et al., 2019*; *Masuda et al., 2019*; *Van Hove et al., 2019*; *Jordão et al., 2019*; *Böttcher et al., 2019*). From these works, it was found that, besides parenchymal microglia, the steady-state CNS also harbors diverse leukocytes localized at the CNS-periphery interfaces, including different subtypes of mononuclear phagocytes (MNPs) such as border-associated macrophages (BAMs), monocytes, and dendritic cells (DCs), along with lymphocytes (T cells, B cells, NK cells, or innate lymphoid cells ILCs) and granulocytes (neutrophils). Several of these immune cell populations have since been shown to play important roles in regulating CNS

development and homeostasis (*Drieu et al., 2022*; *Pasciuto et al., 2020*; *Tanabe and Yamashita, 2018*), or identified as key players in disease models and aging (*Alves de Lima et al., 2020*; *Minhas et al., 2021*). Collectively, these studies have highlighted how understanding vertebrate brain leukocyte heterogeneity is key to describe CNS interactions with the microenvironment and other cells such as glial cells, neurons, or endothelial cells. By contrast, the repertoire of immune cells in the CNS of other vertebrate models remains less well characterized.

This is the case for the zebrafish, an increasingly recognized model for translational research on human neurological diseases, owing to its robust genetics and conserved physiology with mammals (*Turrini et al., 2023*; *Liu, 2023*; *D'Amora et al., 2023*). Zebrafish possess a diverse array of immune cells, encompassing both innate and adaptive lineages. Their innate immune compartment includes well-characterized neutrophils and macrophages, which have been extensively studied in both developmental and disease contexts, as well as dendritic cells and monocytes, which remain comparatively less defined (*Speirs et al., 2024*). In parallel, zebrafish also develop adaptive immune cells such as T and B lymphocytes, which share key molecular and functional features with their mammalian counterparts (*Carradice and Lieschke, 2008*). While immune cells have been described across several organs (*Zhou et al., 2023*; *Wittamer et al., 2011*), a comprehensive characterization of the immune cell populations present in the adult zebrafish brain at steady state is still lacking. This constitutes an essential prerequisite for dissecting the complex cellular orchestration underlying healthy and diseased CNS states. Additionally, although microglia, the CNS-resident macrophages, have been identified in the adult zebrafish brain and profiled using bulk RNA-seq (*Oosterhof et al., 2018*; *Ferrero et al., 2018*; *Ferrero et al., 2021*; *Wu et al., 2020*), and more recently at single-cell resolution (*Zhou et al., 2023*; *Silva et al., 2021*), the extent of phenotypic heterogeneity within the microglial compartment remains unclear. To address this, we established robust protocols for brain dissociation and prospective isolation of leukocytes using fluorescent transgenic lines. By combining this approach with single-cell RNA sequencing, we have generated a gene expression atlas composed of the distinct immune cells present in the homeostatic brain. This dataset revealed the presence of subpopulations of mononuclear phagocytes and other leukocytes, including cell types that have not been or have only been partially characterized so far. Here, we present the characterization of a new mononuclear phagocyte population that represents an important fraction among all brain leukocytes and coexist with microglia in the brain parenchyma. This population of cells is *batf3*-dependent and expresses known DC canonical genes. In light of these observations, we have also revisited the phenotype of myeloid-deficient mutant lines, such as *csf1ra*[-/-], *csf1rb*[-/-], and *irf8*[-/-] fish, that have been instrumental to the field. Overall, we provide an overview of the immune landscape in the adult zebrafish brain which akin to findings in mammals, boasts distinct myeloid and lymphoid cell types.

## Results

### Mononuclear phagocytes represent the main immune cell population in the adult zebrafish brain

As a first step, we sought to assess the leukocytes present in the zebrafish adult brain according to their cellular morphology. We previously showed the *cd45:DsRe*d transgene labels all leukocytes, with the exception of B lymphocytes (*Wittamer et al., 2011*; *Ferrero et al., 2020*). Therefore, we performed May-Grünwald Giemsa (MGG) staining on a pure population of *cd45:DsRed*[+] cells isolated from the brain of adult *Tg(cd45:DsRed)* transgenic animals by flow cytometry (*Figure 1A*). Cells with the classical morphological features of mononuclear phagocytes were identified as macrophages/microglia based on their large and vacuolated cytoplasms (*Figure 1B*), in accordance with our previous work (*Wittamer et al., 2011*). Monocytes, recognized by their kidney-shaped nuclei, were also present, as well as cells with a typical dendritic cell morphology, namely elongated shapes, large dendrites, and oval or kidney-shaped nuclei (*Lugo-Villarino et al., 2010*; *Figure 1B*). We also found large numbers of lymphocytes, clearly distinguished from myeloid cells by their smaller size and narrow and basophilic cytoplasm stained in blue. The remaining cells were neutrophils, characterized by their clear cytoplasm and highly segmented nuclei.

Next, we took advantage of fluorescent zebrafish transgenic lines, allowing to detect and quantify the different leukocyte subsets using flow cytometry. To achieve this, *Tg(cd45:DsRed)* animals were crossed to established GFP reporters that label mononuclear phagocytes (*Tg(mpeg1:GFP)*), neutrophils

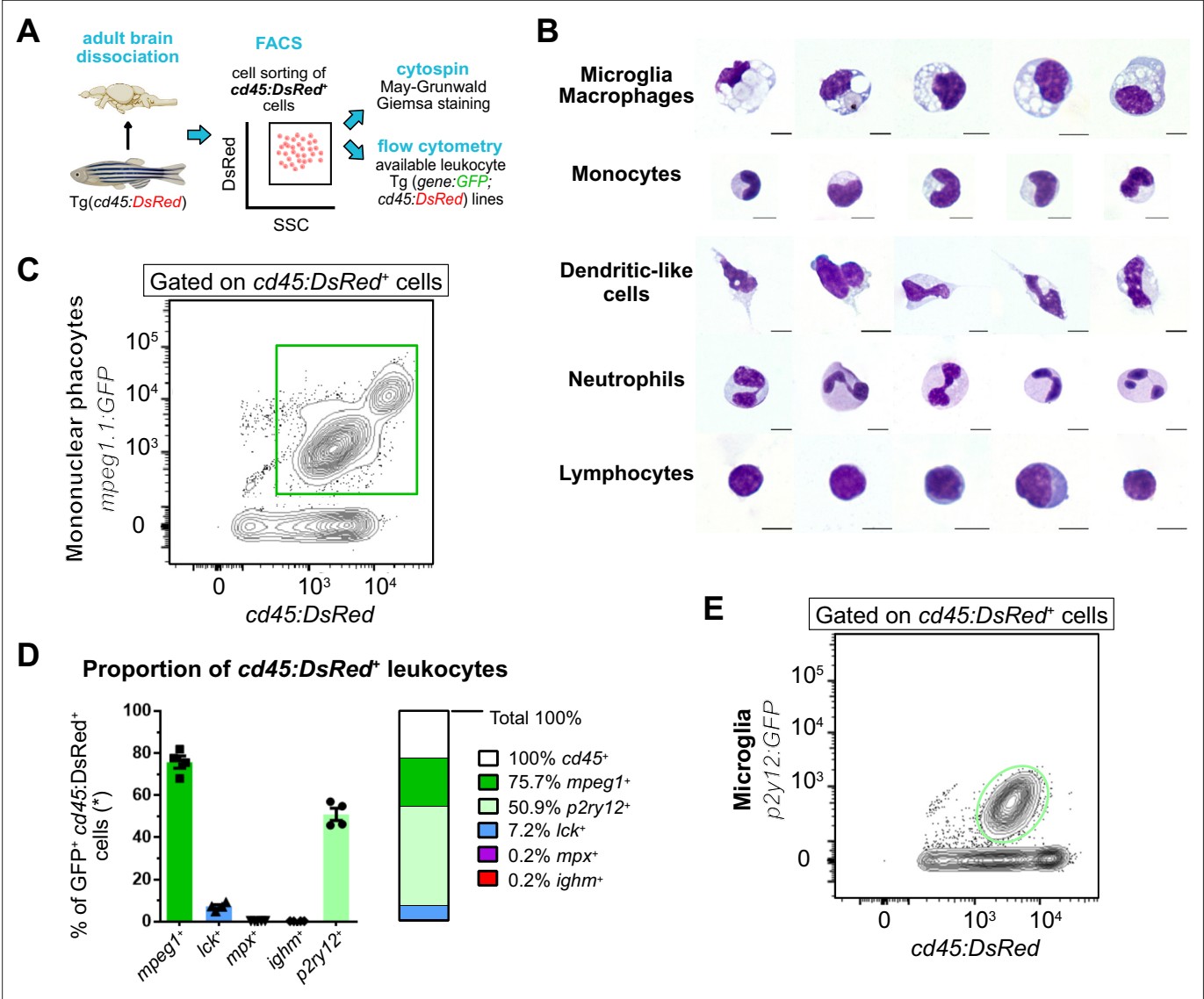

**Figure 1.** Leukocyte heterogeneity in the adult zebrafish brain using blood lineage-specific transgenic lines. (**A**) Schematic overview of the experiment. First, *cd45:DsRed* ⁺ cells were sorted, cytospined, and stained with May-Grunwald Giemsa (MGG). In parallel, lines carrying the *cd45:DsRed* transgene in combination with blood lineage-specific GFP reporters were analyzed by flow cytometry. (**B**) Morphology of brain-sorted *cd45:DsRed* ⁺ cells stained with MGG. Microglia and/or macrophages, monocytes, dendritic cells, neutrophils, and lymphocytes were identified. The scale bar represents 5 μm. C. Flow cytometry analysis on brain cell suspensions from adult *Tg(mpeg1:GFP; cd45:DsRed)* identifying *mpeg1:GFP⁺*; *cd45:DsRed* ⁺ mononuclear phagocytes (green gate). (**D**) Proportion of brain immune cell types, as determined by flow cytometry analysis on cell suspensions from fish carrying *cd45:DsRed* ⁺ and a lineage-specific GFP reporter (n=4 fish). The percentage relative to total *cd45:DsRed⁺* leukocytes is shown, with the exception of *Tg(ighm:GFP; cd45:DsRed)* which are not normalized as the *cd45:DsRed* transgene is not expressed in the B cell lineage. (**E**) Flow cytometry analysis of brain cell suspensions from an adult *Tg(p2ry12:p2ry12-GFP; cd45:DsRed)* fish, identifying *p2ry12:p2ry12-GFP⁺*; *cd45:DsRed⁺* microglial cells (light green gate). n refers to the number of biological replicates. Data in (**D**) are mean ± SEM.

The online version of this article includes the following figure supplement(s) for figure 1:

**Figure supplement 1.** Flow cytometry analyses of brain leukocytes using blood lineage-specific GFP reporter lines.

(*Tg(mpx:GFP)*), NK and T lymphocytes (*Tg(lck:GFP)*), or IgM-expressing B cells (*Tg(ighm:GFP)*) (***Figure 1A***). For each double transgenic line, we quantified by flow cytometry the proportion of GFP⁺ cells within the *cd45:DsRed⁺* population. As expected, we found that *mpeg1*:GFP⁺ mononuclear phagocytes were the most abundant leukocytes in the adult brain, accounting for 75.7±2.9% of the total *cd45:DsRed⁺* population (n=4) (***Figure 1C and D***). In contrast, *mpx:GFP⁺* neutrophils were scarce, representing only 0.2±0.04% of brain leukocytes (n=4) (***Figure 1—figure supplement 1A***).

Regarding lymphocytes, *lck:GFP*[+] NK/T cells were more abundant than *ighm:GFP*[+] cells, accounting for 7.2±0.9% (n=4) and 0.2±0.01% (n=4), respectively (***Figure 1—figure supplement 1B and C***).

Although *mpeg1*-driven fluorescent transgenes are commonly used to label mononuclear phagocytes, we and others have previously shown that *ighm*-expressing B cells are also marked by these reporters, as they endogenously express *mpeg1.1* (***Ferrero et al., 2020***; ***Moyse and Richardson, 2020***). However, based on the low numbers of brain *ighm:GFP*[+] cells identified in our flow cytometry analyses, we estimated their contribution to the *mpeg1*[+] population was minimal and that brain *mpeg1*[+] cells mostly comprise mononuclear phagocytes. To determine the proportion of microglial cells within the broader population of *mpeg1*[+] mononuclear phagocytes, we crossed *Tg(cd45:DsRed)* fish (marking leukocytes), with animals carrying the *Tg(p2ry12:p2ry12-GFP)* transgene (***Sieger et al., 2012***). P2ry12 is a well-established microglia-specific marker conserved across species, including zebrafish (***Mazzolini et al., 2020***; ***Rovira et al., 2023***; ***Ferrero et al., 2018***), and has been previously used to distinguish microglia from other brain mononuclear phagocytes (***Butovsky et al., 2014***). Flow cytometry analysis of brain cell suspensions from double transgenic adults revealed that *p2ry12:GFP*[+] microglia accounted for approximately 51% (±2.9; n=4) of all *cd45:DsRed*[+] leukocytes (***Figure 1D and E***). Meanwhile, mpeg1:GFP[+] ~ 75% of the same cd45:DsRed[+] population. Since the percentage of mpeg1:GFP[+] cells exceed that of p2ry12:GFP[+] microglia, we inferred that roughly 25% of brain mpeg1:GFP[+] mononuclear phagocytes lack p2ry12:GFP transgene expression and are, therefore, likely non-microglial in nature. Indeed, based on our cytological observations, this population likely contains a mixture of monocytes and dendritic cells. Collectively, these analyses suggest an important diversity among leukocytes present in the steady-state brain of the adult zebrafish.

## Single-cell transcriptomics identifies multiple leukocyte populations in the adult brain

To fully characterize the heterogeneity within the zebrafish brain immune landscape, next, we turned to single-cell transcriptome profiling. Viable *cd45:DsRed*[+] cells were FACS-sorted from the steady-state brain of adult *Tg(cd45:DsRed)* animals, then subjected to scRNA-sequencing using the 10 X platform (***Figure 2A***, ***Figure 2—figure supplement 1***). After an unsupervised uniform manifold approximation and projection (UMAP) and single-cell clustering, we obtained a total of 20 cell clusters (***Figure 2B***). A preliminary observation of our dataset, revealed the expression of *cd45* (also known as *ptprc*) in all clusters of the dataset, thus confirming their hematopoietic identity (***Figure 2C***). In addition, expression of canonical genes for mononuclear phagocytes (*mpeg1.1*), neutrophils (*mpx*) or T/NK cells (*lck, lymphocyte-specific protein tyrosine kinase*) were found in several clusters (***Figure 2C***). Together, these initial observations indicated that we were able to capture a repertoire of different brain leukocytes represented in individual cluster identities. This is in line with the cell type diversity determined from our cytological and flow cytometry analyses.

Cluster annotation was achieved based on expression of defined blood lineage-specific genes previously established in zebrafish (***Tang et al., 2017***; ***Hernández et al., 2018***; ***Moore et al., 2016***), and from published transcriptomes from human and mouse brain leukocytes (***Mrdjen et al., 2018***; ***Jordão et al., 2019***; ***Hammond et al., 2019***; ***Masuda et al., 2019***; ***Van Hove et al., 2019***). Using these approaches, we annotated 15 clusters. The remaining cells are included in the online material but were not used for further analysis in this study. We identified seven major leukocyte populations that comprised microglia (MG), macrophages (MF), dendritic-like cells (DC-like), T cells, natural killer cells (NK), innate lymphoid-like cells (ILCs), and neutrophils (Neutro) (***Figure 2B***, ***Supplementary file 1***). Expression of the markers for each cluster is visualized by plotting the top 50 marker genes (***Figure 2D*** and ***Supplementary file 2***). Of note, one cluster was annotated as proliferative (Prolif) because of the expression of proliferative markers, suggesting the presence of dividing brain leukocytes, however, marker genes were not indicative of a specific cell type (***Figure 2B and D*** and ***Supplementary file 1***). A detailed analysis of the different clusters from the lymphoid and myeloid compartments is presented in the following sections, with an emphasis on microglia and DC-like clusters.

## The adult zebrafish brain contains innate and adaptive lymphoid cells

Expression of *lck*, a conserved marker for T lymphocytes and NK cells (***Moore et al., 2016***), identified three clusters of lymphoid cells (***Figure 2C***, ***Supplementary file 1***). Two of them expressed T

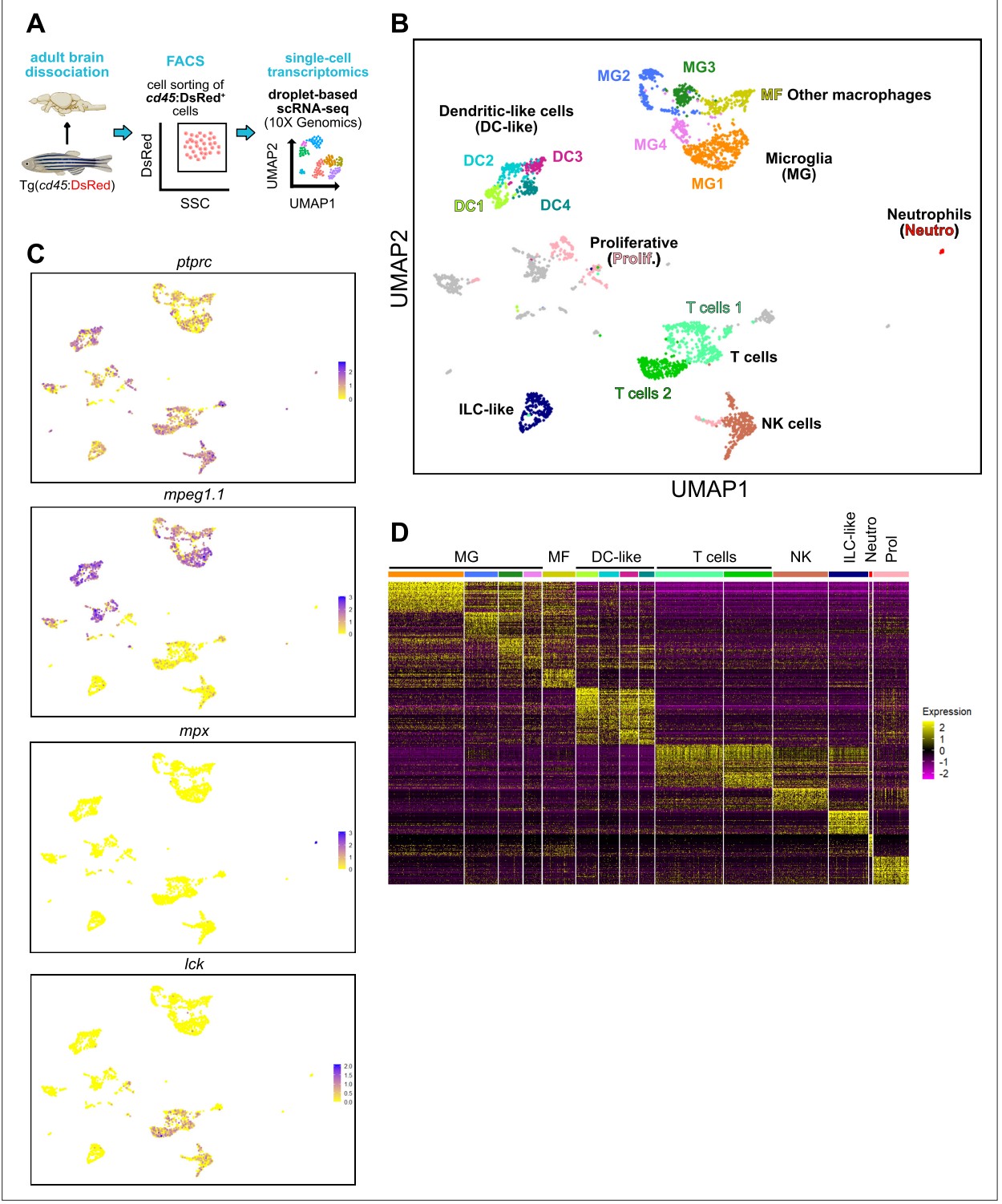

**Figure 2.** Diversity of brain leukocytes as shown by single-cell transcriptomics. (**A**) Schematic overview of the experimental approach. Single-cell profiling of total brain *cd45:DsRed⁺* leukocytes (pool from three individual fish) was performed using the 10 X Genomics platform. (**B**) Split Uniform Manifold Approximation Projection (UMAP) of brain *cd45:DsRed⁺* cells with annotated cell populations. Clusters in gray shade are not indicative of a specific cell type and were not annotated. (**C**) UMAP plots depicting the expression pattern of *ptprc*, also known as *cd45* (leukocytes), *mpeg1.1* (mononuclear phagocytes), *mpx* (neutrophils), and *lck* (T and NK lymphocytes). Gene expression levels from low to high are indicated by a color gradient from yellow to purple (normalized counts in log1p). (**D**) Heat map of the top differentially up-regulated genes in each cluster (row = gene, column = cell type). Color scale (gradual from purple to yellow) indicates the expression level (average log2 fold change).

*Figure 2 continued on next page*

*Figure 2 continued*

The online version of this article includes the following figure supplement(s) for figure 2:

**Figure supplement 1.** Isolation of a pure population of brain leukocytes.

cell-specific marker genes such as *zap70* (tcr-associated protein kinase), TCR co-receptors including *cd4-1*, *cd8a*, *cd8b* and *cd28*, and *il7r*, a cytokine receptor that functions in T cell homeostasis (*Figure 3A and D*), all of them showing conservation between mammals and zebrafish. This suggests that these two *zap70*-expressing clusters contain a mix of CD4$^+$ and CD8$^+$ T cells, and were thus annotated as Tcells1 and Tcells2. Interestingly, a proportion of cells within these clusters expressed *runx3,* which in mammals has been reported as a regulator of tissue resident memory CD8 T cells in different tissues, including the brain (*Milner et al., 2017*). The second cluster highly expressed genes previously described as markers for NK cells in the zebrafish whole kidney marrow (WKM) (*Tang et al., 2017*; *Carmona et al., 2017*), such as chemokines *ccl36.1* and *ccl38.6*, granzymes *gzm3.2* and *gzm3.3*, *il2rb* and *ifng1* (*Figure 3B and D*). However, expression of novel immune-type receptor (*nitr*) or NK-lysin genes was not detected in brain NK cells, in contrast to WKM NK cells (*Carmona et al., 2017*; *Moore et al., 2016*; *Yoder et al., 2010*). Annotation of these lymphoid clusters was mostly based on a zebrafish WKM reference data set (*Tang et al., 2017*) and, therefore, differences may exist between tissues.

Notably, we identified an additional cluster that did not express any of the previously mentioned T cell markers but displayed *il4* and *il13* expression in a large proportion of cells (*Figure 3C and D*). In mammals, these two cytokines identify CD4$^+$ T helper type 2 cells, as well as innate lymphoid cells type 2 (ILC2s), the innate counterparts of adaptive T helper cells. However, unlike T cells, ILC2s lack antigen receptors and associated co-receptors (*Vivier et al., 2018*). Interestingly, this cluster was also positive for *gata3*, a transcription factor that regulates the development and functions of ILC2s (*Wong et al., 2012*). The expression profile identified in this cluster may thus represent the molecular signature of zebrafish ILC2-like cells (*Vivier et al., 2016*). To test this hypothesis, we performed qPCR analyses on *cd45:DsRed$^+$* cells isolated from *rag2*-deficient fish. We hypothesized that, like their murine counterparts (*Spits and Cupedo, 2012*), *rag2* mutant zebrafish, which lack T and B cells (*Tang et al., 2014*), would still produce ILC-like cells. Supporting this postulate, while the expression levels of *lck* and *zap70* was significantly reduced in brain leukocytes from the *rag2* mutants in comparison with that from their *wild-type* siblings (*Figure 3E*), *gata3*, *il4* and *il13* showed similar expression levels between cells from both genotypes (*Figure 3E*). It thus appears that the expression of putative ILC2 cell-associated genes in brain leukocytes is not changed in the absence of T cells. Altogether, these findings support our annotation of this cluster as ILC-like cells.

## Cellular diversity of the mononuclear phagocyte system in the adult zebrafish brain

As shown in *Figure 2C*, expression of *mpeg1.1*, a canonical marker for mononuclear phagocytes, was identified in nine clusters of our dataset. Four clusters were annotated as MG, one as macrophages MF, and four as DC-like cells (*Figure 2B* and *Supplementary file 1*).

MG clusters (MG1, MG2, MG3, MG4) differentially expressed zebrafish microglial genes such as the lipoproteins *apoc1* and *apoeb* (*Herbomel et al., 2001*; *Peri and Nüsslein-Volhard, 2008*; *Ferrero et al., 2018*; *Mazzolini et al., 2020*), ms4a17a.10 (*Oosterhof et al., 2018*) - a member of the membrane-spanning 4 A gene family, galectin 3 binding protein lgals3bpb (*Rovira et al., 2023*; *Kuil et al., 2019*), and hepatitis A virus cellular receptors havcr1 and havcr2 (*Kuil et al., 2019*; *Oosterhof et al., 2018*; *Figure 4A and D*). Moreover, csf1ra and csf1rb, the zebrafish paralogs of CSF1R and well-conserved regulators of microglia development and homeostasis (*Oosterhof et al., 2018*; *Ferrero et al., 2021*; *Hason et al., 2022*), were also identified as marker genes, although their level of expression differed between microglia clusters (*Figure 4D*, *Supplementary file 1*). Importantly, expression of canonical microglial genes were also found in the MG clusters, such as p2ry12, hexb, mertka, and members of the c1q genes, among others, supporting a conserved microglial phenotype (*Butovsky et al., 2014*; *Jurga et al., 2020*; *Butovsky and Weiner, 2018*; *Gerrits et al., 2020*; *Figure 4—figure supplement 1*).

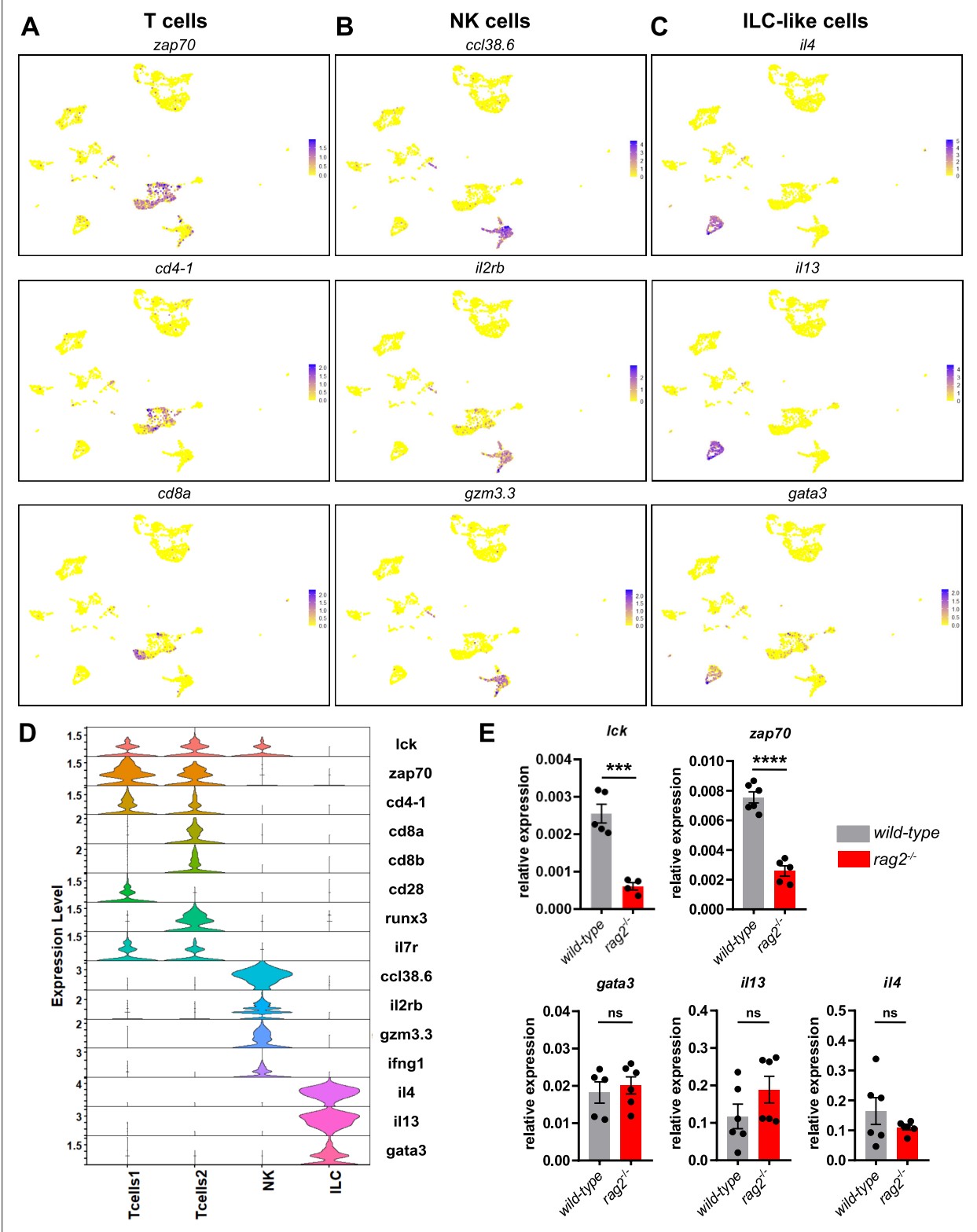

**Figure 3.** Single-cell RNA sequencing identifies several lymphocyte subpopulations in the adult brain. (**A–C**) Uniform Manifold Approximation Projection (UMAP) visualization of the expression of selected genes in the annotated T cell clusters Tcells1 and Tcells2 (*zap70*, *cd4-1*, and *cd8a*), NK cluster (*ccl38.6*, *il2rb*, *gzm3.3*), and ILC-like cluster (*il4*, *il13*, and *gata3*). Color scale (gradual from yellow to purple) indicates the expression level for each gene (normalized counts in log1p). (**D**) Violin plots representing the expression levels of known lymphocyte markers (normalized counts in log1p) within the different clusters. (**E**) Comparison of the relative expression of *lck*, *zap70*, *gata3*, *il13,* and *il4* transcripts between brain *cd45:DsRed* [+] cells isolated by

*Figure 3 continued on next page*

*Figure 3 continued*

FACS from T cell-deficient *rag2*$^{-/-}$ mutants (red bars) and their *wild-type* siblings (gray bars). Each data point represents an individual fish (n=6) and error bars indicate SEM. ***p<0.001, **** *Pp*<0.0001 (Two-tailed unpaired t-test).

We also found a cluster of *mpeg1.1*-expressing cells that we annotated as *non-microglia macrophages* (MF). Similar to the microglia clusters (MG), this cluster differentially expressed macrophage-related genes such as *marco*, *mfap4*, *csf1ra*, and components of the complement system (e.g. *c1qb*) (***Figure 4B and D***, ***Supplementary file 1***). However, this cluster differed from the four microglia clusters because microglia markers were not found. This cluster also showed high expression of calcium-binding proteins such as *s100a10b*, *anxa5b*, and *icn*, as well as the coagulation factor XIII *f13a1b*, among others (***Figure 4B and D*** and ***Supplementary file 1***). In contrast to mammals, the distinction between microglia and other macrophages in the adult zebrafish brain (i.e. border-associated macrophages) is still unclear (***Silva et al., 2021***) and to date, no known marker or fluorescent reporter line is available to distinguish these two related cell types. Another possibility is that these *mpeg1.1*-expressing cells are blood-derived monocytes/macrophages. In order to better characterize these two *mpeg1.1*-expressing clusters, we performed a differential expression analysis between MF and MG (all four clusters together). As shown in ***Figure 4E***, microglial genes such as *apoeb*, *apoc1*, *lgals3bpb*, *ccl34b.1*, *havcr1*, and *csf1rb* were significantly down-regulated, whereas macrophage-related genes such as *s100a10b*, *sftpbb*, *icn*, *fthl27*, *anxa5b*, *f13a1b* and *spi1b* were significantly up-regulated (***Supplementary file 3***). Therefore, these genes may thus serve as novel markers to discriminate these two related types of macrophages.

Finally, our analysis identified a third group of *mpeg1.1*-expressing cells represented in four clusters (DC1, DC2, DC3, DC4) (***Figure 2B***). Highly expressed genes in these clusters included *siglec15l* (sialic acid binding Ig-like lectin 15, like) and *ccl19a.1* (C-C motif ligand 19 a), a putative ligand of the zebrafish T cell receptor *ccr7* (***Wu et al., 2012***; ***Figure 4C and D*** and ***Supplementary file 1***). Intriguingly, these four clusters expressed *id2a*, *xcr1a.1*, *batf3* (basic leucine zipper ATF-like 3 transcription factor), and *flt3* (***Figure 4C and D*** and ***Supplementary file 1***), which are the orthologs of the mammalian Id2a, Xcr1, Batf3, and Flt3 genes, required for development and/or functions of conventional dendritic cells (cDC1) (***Cabeza-Cabrerizo et al., 2021***). These clusters also expressed *chl1a* (adhesion molecule L1), reported to promote DC migration through endothelial cells (***Maddaluno et al., 2009***), and *hepacam2* (***Figure 4D***, ***Supplementary file 1***), frequently found in mammalian DC expression datasets. However, all four clusters had negligible expression of any of the microglia or macrophage markers previously mentioned (***Figure 4D***, ***Supplementary file 1***). Based on their transcription profile and possible shared characteristics with mammalian DCs, these clusters were annotated as DC-like cells (DC1, DC2, DC3, DC4).

We next conducted a differential expression analysis of DC-like cells (DC1, DC2, DC3, DC4) versus MG (MG1, MG2, MG3, MG4), as two separate clusters. As shown in ***Figure 4F***, significantly different genes include genes previously found as DC-like (up-regulated) or microglial (down-regulated) markers, thus confirming their distinct transcriptomic profiles. In addition, DC-like cells could also be identified based on differential expression of *irf8*, *ptprc*, and *mpeg1.1*, all significantly up-regulated in this population in comparison to MG (***Figure 4F***, ***Supplementary file 3***). This is similar to mammalian cDC1, which are IRF8[high], PTPRC (CD45)[high], and MPEG1[high] (***Cabeza-Cabrerizo et al., 2021***), and thus strengthens the idea that DC-like cells phenotypically resemble mammalian cDC1. In order to explore the biological function of MG and DC-like cells, we performed pathway enrichment analysis (using GO Biological Processes and Reactome) for each MG and DC-like markers (***Supplementary file 4*** and see Materials and methods). This analysis enriched for terms in MG such as *endosomal lumen acidification* (e.g. H+ ATPase family genes), *synapse pruning* (e.g. C1QC/*c1qc*), *response to lipoprotein particle* (e.g. ABCA1/*abca1b*, APOE/*apoeb*), *interleukin-10 signalling* (e.g. IL10RA/*il10ra*), *macrophage activation* (e.g. CTSC/*ctsc*, HAVCR2/*havcr2*), *MHC class II antigen presentation* (e.g. CD74/*cd74a*, HLA-DOB/*mhc2b*), *complement cascade* (e.g. C1QA/*c1qa*, CFP/*cfp*), *mononuclear cell migration* (e.g. CSF1R/*csf1rb*, CMKLR1/*cmklr1*), or *phagocytosis* (e.g. MERTK/*merkta*, MARCO/*marco*) (***Figure 4—figure supplement 2A***, ***Supplementary file 4*** and see Materials and methods). Enriched terms in DC-like included *FLT3 signaling* (e.g. FLT3/*flt3*), *myeloid cell differentiation* (e.g. BATF3/*batf3*, ID2/*id2a*), *Rac2 GTPase cycle* (e.g. RAC2/*rac2*, CDC42/*cdc42l*), *Fc receptor signaling pathway* (e.g. FCER1G/*fcer1g*), *cell chemotaxis* (e.g. CCL19/*ccl19a.1*, XCR1/*xcr1a.1*), innate signaling

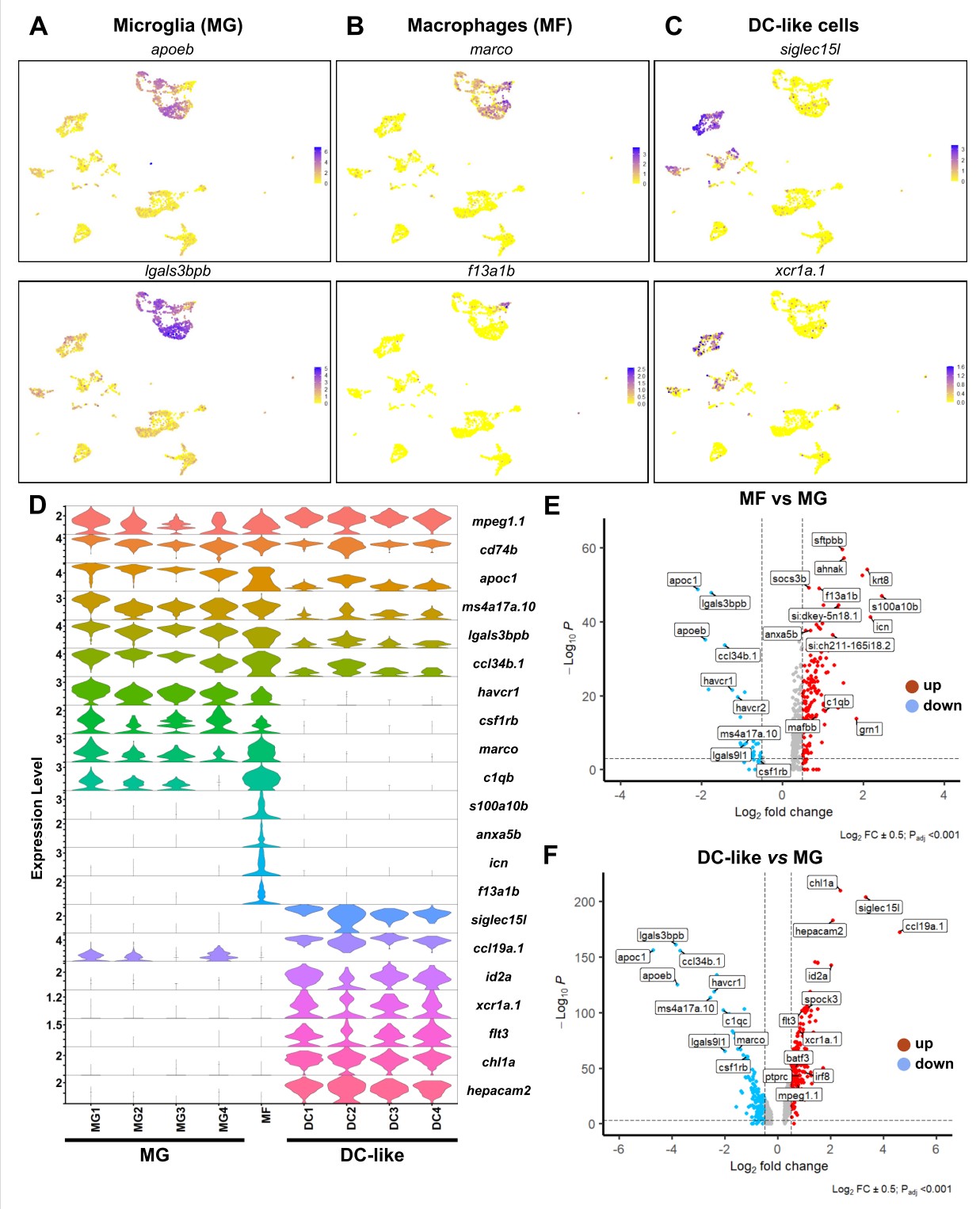

**Figure 4.** Heterogeneous subsets of mononuclear phagocytes exist in the zebrafish brain. (**A–C**) Uniform Manifold Approximation Projection (UMAP) visualization of the expression of selected genes in the microglia (*apoeb* and *lgals3bpb*) (**A**), non-microglia macrophage (*marco* and *f13a1b*), (**B**) and DC-like (*xcr1a.1* and *siglec15l*), (**C**) cell clusters. Color scale (gradual from yellow to purple) indicates the expression level for each gene (normalized counts in log1p). (**D**) Violin plot analysis comparing the expression levels of selected genes (y-axis, normalized counts in log1p) between the different mononuclear phagocyte cell clusters. (**E**) Volcano plot showing the differentially expressed (DE) genes between microglia (MG) and non-microglia macrophages (MF). Lines indicate significantly DE genes (log2 fold-change >|0.5|, -log10 $P_{adj}$ <0.001). Red dots represent up-regulated genes and blue

*Figure 4 continued on next page*

*Figure 4 continued*

dots down-regulated genes. Labels show representative DE genes identified in the analysis. (F) Volcano plot showing the DE genes between MG and DC-like cells. Lines indicate significantly DE genes (log2 fold-change >|0.5|, -log10 P$_{adj}$ <0.001).

The online version of this article includes the following figure supplement(s) for figure 4:

**Figure supplement 1.** Canonical microglial genes conserved between zebrafish and mammals.

**Figure supplement 2.** Functional analysis using the corresponding mammalian orthologs.

pathways, such as *toll-like receptor cascades* (e.g. TLR6/*tlr1*, IRAK3/*irak3*) as well as terms involved in adaptive immunity, such as *alpha-beta T cell activation* (e.g. CBLB/*cblb*, SOCS1/*socs1*) or *lympho-cyte activation involved in immune response* (e.g. IL12B/*il12ba*) (*Figure 4—figure supplement 2B*). Moreover, we used the Enrichr tool to predict the annotation of the MG and DC-like clusters using the PanglaoDB database that contains multiple single-cell RNA sequencing experiments from mouse and human (*Franzén et al., 2019*). The three top significant cell types for MG marker genes were 'microglia,' 'monocytes,'and 'macrophages,' while for DC-like were 'Dendritic Cells,' 'Plasmacytoid DCs,' and 'Langerhans Cells' (*Figure 4—figure supplement 2C and D*).

## Two phenotypically distinct populations of mpeg1⁺ cells within the brain parenchyma

Having demonstrated the diversity of the immune landscape of the adult zebrafish brain, we next sought to investigate the tissue localization of the different leukocyte populations identified in our dataset, using the same transgenic lines as in *Figure 1*. To differentiate microglia from the two pheno-typically distinct populations of brain mononuclear phagocytes (MF and DC-like), we first examined adult brain sections of *Tg(mpeg1:GFP)* and *Tg(p2ry12:p2ry12-GFP)* single transgenic fish immunola-beled for GFP and the pan-leukocytic marker L-plastin (Lcp1). We found the majority of L-plastin⁺ cells within the brain parenchyma co-expressed the *mpeg1:GFP* transgene (*Figure 5A–C*). Upon examina-tion of *Tg(p2ry12:p2ry12-GFP)* fish, however, we observed that not all parenchymal L-plastin⁺ cells were GFP (*Figure 5D–F*). Analysis of *Tg(p2ry12:p2ry12-GFP; mpeg1:mCherry)* double transgenics confirmed these observations, a.k.a. that a fraction of *mpeg1:mCherry*⁺ cells was negative for the microglial *p2ry12:p2ry12-GFP* transgene (*Figure 5G–J*). This suggested that non-microglia mpeg1-expressing cells are present in the brain parenchyma. Interestingly, in contrast to GFP⁺; mCherry⁺ microglia which are abundant across brain regions, GFP⁻; mCherry⁺ cells particularly localized in the ventral part of the posterior brain (midbrain and hindbrain) (*Figure 5G–J*). Notably, these cells presented with a highly branched morphology when compared to GFP⁺; mCherry⁺ microglia.

Based on these findings, we next investigated brain samples from *Tg(mhc2dab:GFP; cd45:DsRed)* fish, where co-expression of both fluorescent reporters specifically labels mononuclear phagocytes (*Wittamer et al., 2011*; *Ferrero et al., 2018*). In our previous work, we had already observed that, in the brain of these animals, two phenotypically distinct cell populations could be isolated by flow cytometry based on differential *cd45:DsRed* expression levels. While the *cd45^low^; mhc2⁺* fraction was clearly identified as microglia due to their specific expression of *apoeb* and *p2ry12*, the exact identity of the *cd45^high^; mhc2⁺* cells remained unclear. However, we initially found these cells lack expression of *csf1ra* transcripts (*Ferrero et al., 2018*) which, in light of our single-cell transcriptomic data, excluded them as macrophages and point to a DC-like cell identity. So, to evaluate the tissue localization of *cd45^high^; mhc2⁺* cells, we performed direct imaging of transgene fluorescence on vibratome brain sections from *Tg(mhc2dab:GFP; cd45:DsRed)* fish (*Figure 5K*). Most GFP⁺ cells were DsRed negative, suggesting the low expression of the *cd45* transgene in microglia likely precluded direct imaging of DsRed in these cells. However, in the ventral part of the posterior brain (midbrain and hindbrain), we observed a clear population of GFP⁺; DsRed⁺ cells, with a highly ramified morphology (*Figure 5K-N*, *Figure 5—figure supplement 1*). The reliable detection of endogeneous DsRed signal in these cells likely identified them as DsRed^high^. Altogether, these observations strongly suggested that the *mpeg1⁺; p2ry12⁻* and *cd45^high^; mhc2⁺* cells represent the same parenchymal, non-microglial population, possibly corresponding to the putative DC-like cells identified in our single-cell transcriptomic dataset.

Finally, we also examined the localization of neutrophils and lymphoid cells, labeled using the *Tg(mpx:GFP)*, *Tg(lck:GFP)*, and *Tg(ighm:GFP)* lines, respectively (*Figure 5—figure supplement 1*). In accordance with *mpeg1⁺; Lcp1⁺* cells being the main leukocyte population present in the adult

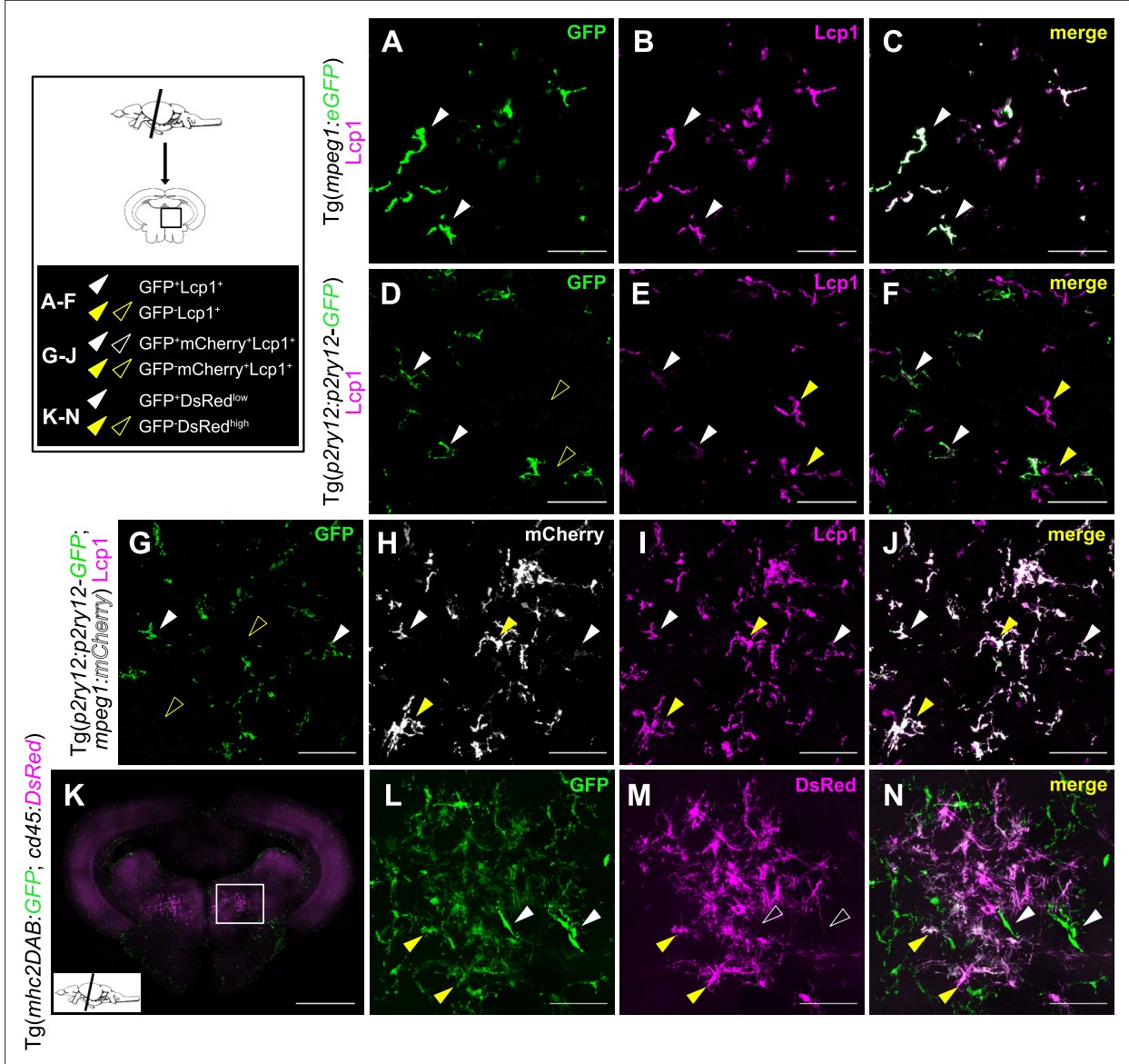

**Figure 5.** Dendritic cell (DC)-like cells localize together with microglia within the brain parenchyma. (**A–F**) Immunofluorescence on transversal brain sections (14 μm) from *Tg(mpeg1:GFP)* (**A–C**) or *Tg(p2ry12:p2ry12- GFP)* (**D–F**) transgenic adult fish co-immunostained with anti-GFP (green) and anti-Lcp1 (magenta) antibodies. (**A–C**) All *mpeg1*:GFP+ mononuclear phagocytes in the brain parenchyma display Lcp1 immunostaining, as expected. (**D–F**) Similarly, all microglial cells, identified by GFP expression in the brain parenchyma of *Tg(pr2y12:p2ry12-GFP)* fish, are Lcp1+, as expected. (**G–J**) In sections of adult *Tg(p2ry12:p2ry12-GFP; mpeg1:mCherry)* double transgenic animals, GFP labeling is not observed in all mCherry+ cells. GFP (green), mCherry (gray), Lcp1 (magenta), and merge of the three channels. All images were taken using a 20 X objective and correspond to orthogonal projections. White arrowheads point to microglial cells (GFP+; Lcp1+or GFP+; mCherry+; Lcp1+) and yellow arrowheads to DC-like cells (GFP-; Lcp1+ or GFP-; mCherry+; Lcp1+). Scale bars: 50 μm. (**K–N**) Confocal imaging of a midbrain vibratome section (100 μm) from an adult *Tg(mhc2dab:GFP; cd45:DsRed)* brain. GFP (green), DsRed (magenta) and merge of the two channels are shown. Images correspond to orthogonal projections, white arrowheads point to GFP+; DsRed+ cells, and yellow arrowheads to GFP-; DsRedhigh. Scale bar in (**K**): 500 μm, scale bar in (**L–N**): 50 μm. Images are representative of brain tissue sections from 2 to 3 fish.

The online version of this article includes the following figure supplement(s) for figure 5:

**Figure supplement 1.** Distribution of dendritic cell (DC)-like cells and immunofluorescence staining for neutrophils and lymphoid cells in the adult brain.

zebrafish brain parenchyma and with our previous flow cytometry analysis, *mpx⁺*, *lck⁺*, and *ighm⁺* cells were rarely found and, if present, they were located at the border of the sections or lining the ventricles (*Figure 5—figure supplement 1*).

Collectively, our findings demonstrated that the adult zebrafish brain parenchyma contains at least two phenotypically distinct populations of mononuclear phagocytes: microglia, and a population of highly branched cells with restricted spatial distribution. Notably, these two populations can be reliably distinguished using a combination of existing transgenic lines.

## Transcriptomic analyses identify DC-like cells as a parenchymal population along with microglia

To assess whether brain *mpeg1:mCherry⁺; p2ry12:GFP⁻* and *cd45:DsRed^high; mhc2:GFP⁺* cells do indeed represent a population distinct from microglia, we next performed bulk transcriptomic analyses to compare their expression profiles. As a source for these studies, we used both *Tg(p2ry12::p2ry12-GFP; cd45:DsRed)* and *Tg(mhc2dab:GFP; cd45:DsRed)* adult fish, allowing for FACS-sort microglia identified in these animals as GFP⁺; DsRed⁺ or GFP⁺; DsRed^low cells, respectively (*Ferrero et al., 2018*). The second population of interest was obtained using the *Tg(mhc2dab:GFP; cd45:DsRed)* reporter, and isolated as GFP⁺; DsRed^high cells (*Figure 6A–C*).

Differential expression analysis between *mhc2dab:GFP⁺; cd45:DsRed^high* (or putative DC-like cells) and *p2ry12:GFP⁺; cd45:DsRed⁺* (or microglia) cells showed up-regulation of DC-like genes previously found in our single-cell transcriptomic analysis (*Figure 6D*, *Supplementary file 5*). Similar results were obtained when comparing DC-like cells with microglia FACS-sorted as *mhcdab⁺; cd45⁺* cells (*Figure 6E*, *Supplementary file 5*). These analyses confirm that our annotated DC-like cluster and *cd45:DsRed^high; mhc2dab:GFP⁺* cells share a similar transcriptome distinct from microglia.

Interestingly, a previous study reported the presence of two heterogeneous populations of *mpeg1*-expressing cells in the adult zebrafish brain. These cells, which were annotated as *phagocytic* and *regulatory* microglia, could be discriminated based on differential expression of the *ccl34b.1:GFP* reporter (*Wu et al., 2020*). Interestingly, these two populations displayed a similar morphology, neuroanatomical location, and differential gene expression pattern than the annotated DC-like and microglia populations identified in our dataset. We thus re-analyzed the data from Wu et al. Differential expression between *regulatory* (*ccl34b.1⁻; mpeg1⁺*) and *phagocytic* (*ccl34b.1⁺; mpeg1⁺*) cells demonstrated up-regulation of genes such as *siglec15l*, *spock3*, *chl1a*, *flt3*, *hepacam2*, *ccl19a.1*, *id2a* and *epdl1*, and down-regulation of genes such as *p2ry12*, *ccl34b.1*, *apoeb*, *apoc1*, *lgals3bpb*, *lgals9l1*, and *havcr1*, among others (*Figure 6—figure supplement 1A* and *Supplementary file 5*). Notably, a large proportion of these DE genes overlapped with that previously found when comparing *mhc2dab:GFP⁺; cd45:DsRed^high* DC-like cells and *p2ry12:GFP⁺; cd45:DsRed⁺* microglia (*Figure 6—figure supplement 1B and C* and *Supplementary file 5*). B cell-related genes such as *ighz* and *pax* were up-regulated (*Figure 6—figure supplement 1A*), suggesting the presence of B cells in the *ccl34b.1⁻; mpeg1⁺* fraction, as expected (*Ferrero et al., 2020*; *Moyse and Richardson, 2020*). In addition, the expression profile of *ccl34b.1⁺; mpeg1⁺* phagocytic microglia strongly correlated with that of *p2ry12:GFP⁺; cd45:DsRed⁺*; and *mhc2dab:GFP⁺; cd45:DsRed⁺* microglia (0.76 and 0.71, respectively), whereas *ccl34b.1⁻; mpeg1⁺* regulatory microglia correlated with *mhc2dab:GFP⁺; cd45:DsRed^high* DC-like cells (0.57) (*Figure 6—figure supplement 1D and E* and *Supplementary file 5*). Collectively, these findings suggest that, at the transcriptomic level, *ccl34b.1⁺; mpeg1.1⁺* cells correspond to microglia in our dataset, and *ccl34b.1⁻; mpeg1⁺* cells resemble the population we annotated as DC-like cells.

## Brain parenchymal DC-like cells are *batf3*-dependent

Our results so far suggested the existence of a putative DC-like cell population located in the parenchyma of the healthy zebrafish brain. To strengthen our findings, we next developed a strategy to assess the identity of this population. We reasoned that the development of the zebrafish counterparts of mammalian cDC1 would likely rely on a conserved genetic program. In our single-cell transcriptomic analysis, zebrafish DC-like cells expressed *batf3*, a cDC1-required transcription factor in human and mouse (*Cabeza-Cabrerizo et al., 2021*). Therefore, using CRISPR/Cas9 technology, we generated a zebrafish *batf3* mutant as a model to explore the lineage identity of putative zebrafish DC-like cells. This mutant line carries an 8 bp deletion downstream of the ATG start, leading to a frameshift mutation and the generation of three premature stop codons (*Figure 7—figure supplement 1*). The resulting

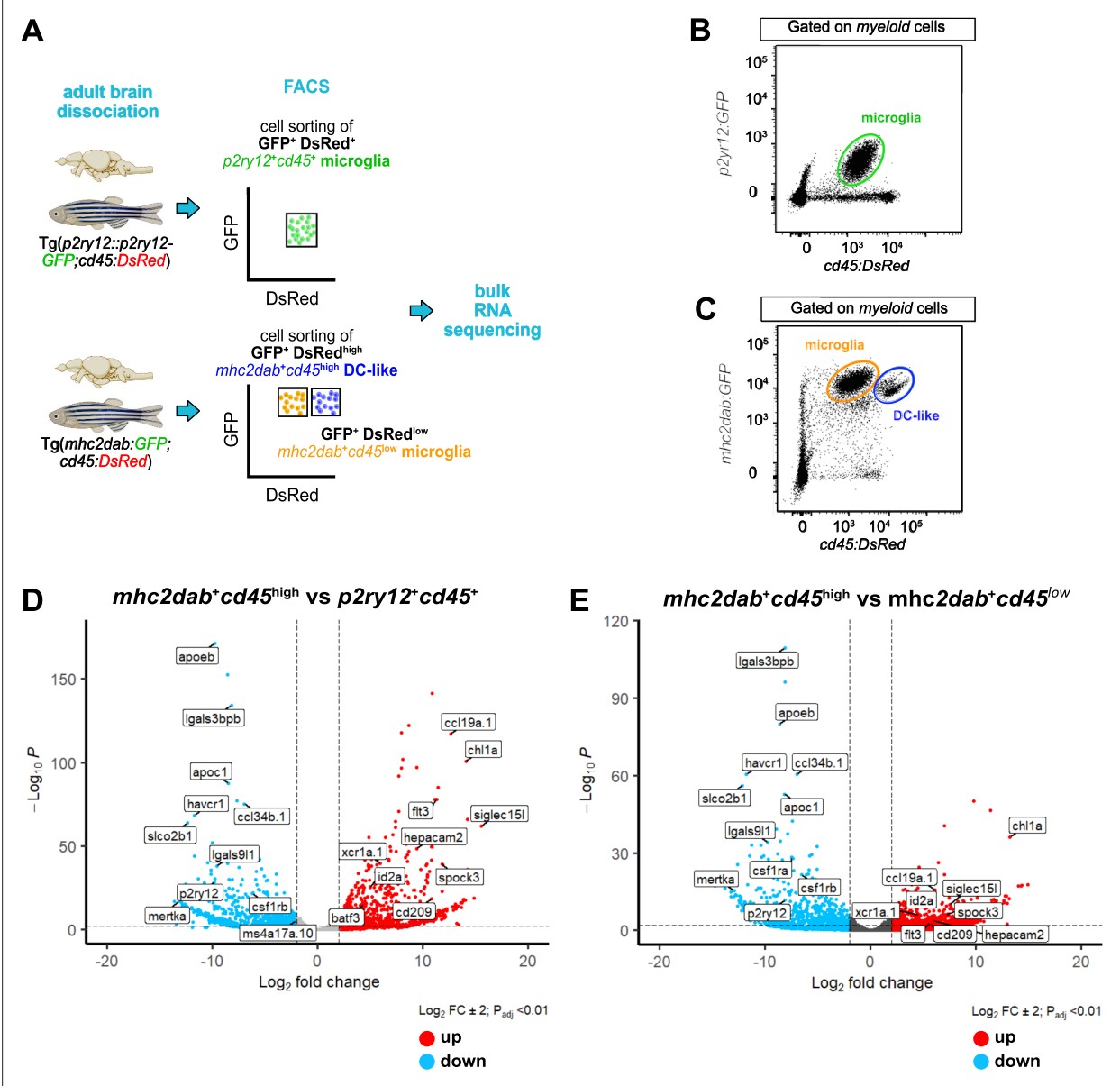

**Figure 6.** Transcriptomic analysis of microglia (*p2ry12+*; *cd45+* or *mhc2dab+*; *cd45low*) and dendritic cell (DC)-like cells (*mhc2dab+*; *cd45high*). (**A**) Schematic overview of the experiments. Microglia were isolated using *Tg(p2ry12:p2ry12-GFP; cd45:DsRed)* or *Tg(mhc2dab:GFP; cd45:DsRed)* transgenic fish, and DC-like cells using the *Tg(mhc2dab:GFP; cd45:DsRed)* reporter line. (**B**) Representative flow cytometry plot identifying microglial cells in brain cell suspensions from *Tg(p2ry12:p2ry12-GFP; cd45:DsRed)* fish. (**C**) Representative flow cytometry plot identifying *mhc2dab:GFP+*; *cd45:DsRedlow* microglia from *mhc2dab:GFP+*; *cd45:DsRedhigh* DC-like cells in brain cell suspensions from *Tg(mhc2dab:GFP; cd45:DsRed)* fish. (**D**) Volcano plot showing the differentially expressed (DE) genes between *mhc2dab+*; *cd45high* DC-like cells and *p2ry12+*; *cd45+* microglia. Red dots represent up-regulated genes and blue dots represent down-regulated genes. Lines indicate significantly DE genes (log2 fold-change >|2|, -log10 $P_{adj}$ <0.01). Labels show marker genes for DC-like cells and microglia identified in the scRNA-sequencing analysis. (**E**) Volcano plot showing the DE genes between *mhc2dab+*; *cd45high* DC-like cells (blue) and *mhc2dab +cd45* low microglia (red). Lines indicate significantly DE genes (log2 fold-change >|2|, -log10 $P_{adj}$ <0.01).

The online version of this article includes the following figure supplement(s) for figure 6:

**Figure supplement 1.** Differential expression analysis reveals that brain *ccl34b.1:GFP+*; *mpeg1.1:mCherry +* cells have a dendritic cell (DC)-like transcriptome.

protein lacks the DNA-binding and basic-leucine zipper domains and is likely to be non-functional. To evaluate whether brain DC-like cells were present in these animals, we crossed the *batf3* mutant line to *Tg(p2ry12:p2ry12-GFP; mpeg1:mCherry)* double transgenic fish, and performed immunostainings of adult brain sections (***Figure 7A–J***). Because DC-like cells are abundant in the ventral posterior

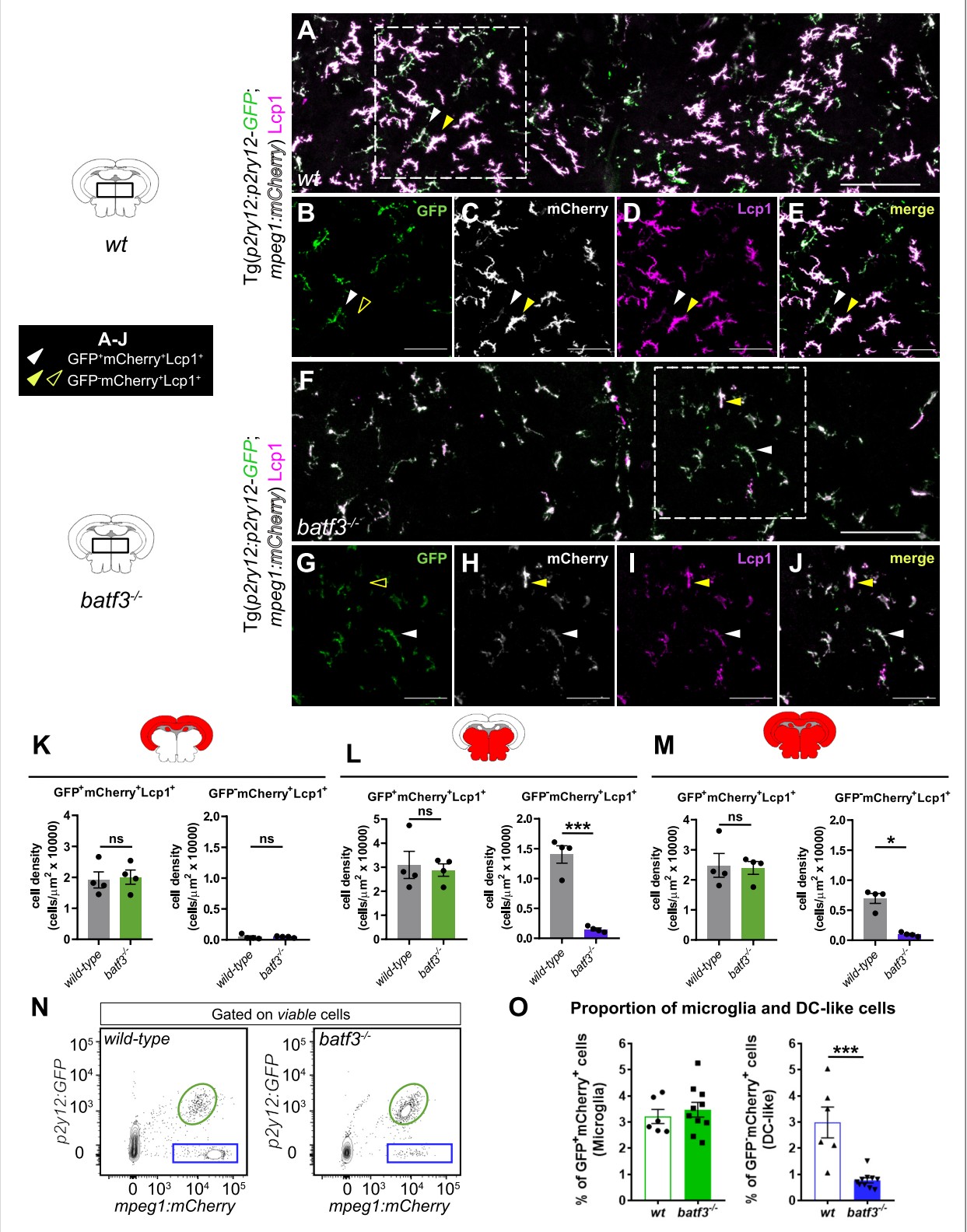

**Figure 7.** Brain dendritic cell (DC)-like cells are lost in *batf3⁻/⁻* mutant fish. (A–J) Immunofluorescence on transverse brain sections (14 μm) from adult *wild-type* (**A–E**) and *batf3⁻/⁻* mutant (**F–J**) fish carrying the *Tg(p2ry12:p2ry12-GFP; mpeg1:mCherry)* double transgene and immunostained for GFP (green), mCherry (gray), and Lcp1 (magenta). Illustrative case of the merge of the three channels (**A, F**) allowing to identify GFP⁺; mCherry⁺; Lcp1⁺ microglia (white arrowheads) versus GFP⁻; mCherry⁺; Lcp1⁺ DC-like cells (yellow arrowheads). While DC-like cells are found in high numbers within

*Figure 7 continued on next page*

*Figure 7 continued*

the ventral part of control parenchyma (**A**), these are dramatically decreased following genetic loss of *batf3* (**F**). Scale bars: 100 μm. (**B–E, G–J**). Single channels high magnification of the insets in A (**B–E**) and F (**G–J**). Scale bars: 50 μm. Images were taken using a 20 X objective and correspond to orthogonal projections. (**K–M**) Quantification of cell density for GFP+; mCherry+; Lcp1+ microglia and GFP-; mCherry+; Lcp1+ DC-like cells in the dorsal midbrain area or optic tectum (**K**), ventral midbrain area (**L**), and the entire section (**M**) of control (gray bars) and *batf3*-/- (green bars) fish. Each dot represents a single fish and data are mean ± SEM. \*p<0.05 (Mann-Whitney test), \*\*\*p<0.0001 (Two-tailed unpaired t-test). (**N**) Flow cytometry analysis of brain cell suspensions from *wild-type* and *batf3*-/- adult fish carrying the *Tg(p2ry12:p2ry12-GFP; mpeg1:mCherry)* reporter. The GFP+; mCherry+ fraction identifies microglia (green circle), whereas the GFP-; mCherry+ fraction contains mainly DC-like cells (blue frame). (**O**) Percentage of microglia and DC-like cells in brain cell suspensions for each genotype, relative to the whole living brain population, as shown in (**N**) (*wild-type*, n=6; *batf3*-/-, n=10). \*\*\*p<0.001 (Two-tailed unpaired t-test). *n* refers to number of biological replicates.

The online version of this article includes the following figure supplement(s) for figure 7:

**Figure supplement 1.** Generation of *batf3*-/- CRISPR mutants.

**Figure supplement 2.** Characterization of brain immune cells in the *batf3*-/- mutant.

brain, we quantified the dorsal (mostly containing the optic tectum) and ventral areas separately, as well as the whole section. The numbers of GFP+; mCherry+; Lcp1+ microglia were similar to their *wild-type* siblings, whereas the ventral posterior brain of homozygous *batf3* mutants was largely devoid of GFP-; mCherry+; Lcp1+ cells, which identify DC-like cells in our model (*Figure 7K–M*). Moreover, we did not observe any changes in the density of other brain leukocytes (*Figure 7—figure supplement 2A–C*). Flow cytometry analyses of brain cell suspensions confirmed the dramatic loss of GFP-; mCherry+ cells in the absence of *batf3* (2.98±0.588%, n=6 vs 0.77±0.097%, n=10) (*Figure 7N and O*). Notably, expression of DC-like markers was barely detectable in the remaining GFP-; mCherry+ cells (*Figure 7—figure supplement 2D and E*). However, these cells also displayed lower mCherry signal intensity, suggesting they most likely represent non-parenchymal *mpeg1*-expressing MF or B cells (*Figure 7—figure supplement 2F and G*). Regarding GFP+; mCherry+ microglia, their proportion was unchanged when compared to that of control fish (*Figure 7N and O*), which is concordant with our initial observations. Finally, we also performed direct imaging of transgene fluorescence on vibratome brain sections of *batf3* mutants carrying the *cd45:DsRed* transgene. In line with our observations, we found that loss of function of *batf3* in *Tg(cd45:DsRed)* transgenic fish resulted in the complete absence of DsRed^high DC-like cells in the ventral area of the midbrain parenchyma in comparison with control brains (*Figure 7—figure supplement 2H–K*). Collectively, these results demonstrated that the population we annotated as DC-like cells is *batf3*-dependent, similar to mammalian cDC1. These results reinforced our hypothesis that these cells represent the zebrafish counterparts of mammalian cDC1.

## Characterization of microglia and dendritic-like cells in mononuclear phagocyte-deficient mutants

The presence of two distinct mononuclear phagocyte subsets in the brain parenchyma made us wondered about their respective status in commonly used microglia-deficient zebrafish lines, as they were all initially characterized using the pan-mononuclear phagocyte *Tg(mpeg1:GFP)* reporter (*Oosterhof et al., 2018*; *Ferrero et al., 2021*; *Wu et al., 2020*). With the ability to discriminate between both populations of microglia and DC-like cells, we thus next sought to examine in more detail the phenotype of the *irf8*-/-, *csf1ra*-/-, *csf1rb*-/- and *csf1ra*-/-; *csf1rb*-/- double mutant (*csf1r^DM*) alleles. To do so, we crossed each mutant line to *Tg(p2ry12:p2ry12-GFP)* animals and analyzed brain sections costained for GFP and L-plastin (*Figure 8*). According to our model, in this setup, microglia will be labeled as GFP+; Lcp1+, while GFP-; Lcp1+ cells will mostly include DC-like cells, easily identified based on their typical ramified cell shape (*Figure 8A–D*). In addition to DC-like cells, GFP-; Lcp1+ cells may also include lymphocytes and/or neutrophils, which are anyway, in much lower numbers than mononuclear phagocytes in the adult brain (*Figure 1D*).

IRF8 is a transcription factor essential for the development of mononuclear phagocytes in vertebrates (*Yáñez and Goodridge, 2016*), including zebrafish (*Ferrero et al., 2020*), where absence of *irf8* results in lack of microglia (*Shiau et al., 2015*; *Earley et al., 2018*). In line with these findings, we found that adult *irf8* homozygous displayed a dramatic, albeit not complete, reduction of GFP+ microglial cells (*Figure 8E–H, U and W*). Interestingly, most remaining microglia localized near or along the ventricle borders, and exhibited characteristics reminiscent of an immature phenotype, e.g.,

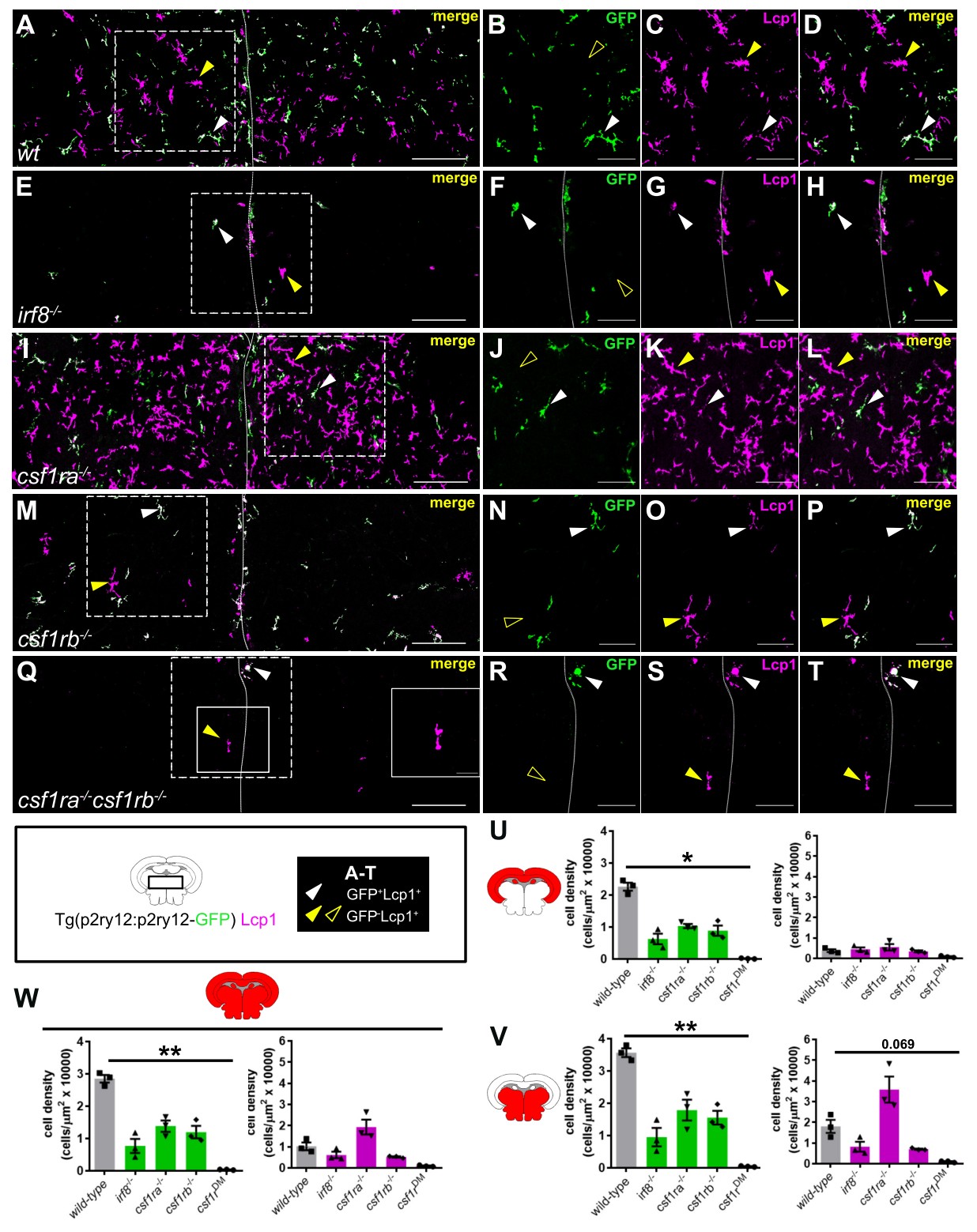

**Figure 8.** Examination of microglia and dendritic cell (DC)-like cells in myeloid–deficient mutant lines. (**A–D**) Immunofluorescence on transverse brain sections from *Tg(p2ry12:p2ry12-GFP)* transgenic adult *wild-type* (**A– D**), *irf8⁻ᐟ⁻* (**E–H**), *csf1ra⁻ᐟ⁻* (**I–L**), *csf1rb⁻ᐟ⁻* (**M–P**) and *csf1ra⁻ᐟ⁻; csf1rb⁻ᐟ⁻ (csf1rᴰᴹ)* (**Q–T**) fish, co-stained with anti-GFP (green) and Lcp1 (magenta) antibodies. (**A, E, I, M, Q**) For each genotype, illustrative case of the merge of the two channels, allowing to discriminate in the parenchyma GFP⁺; Lcp1⁺ microglia (white arrowheads) from GFP⁻; Lcp1⁺ DC-like cells (yellow arrowheads). Single channels high magnification of the insets (dashed frame) in A (**B–D**), E (**F–H**), I (**J–L**), M (**N–P**), and Q (**R–T**). Outline yellow arrowheads indicate the absence of GFP

*Figure 8 continued on next page*

*Figure 8 continued*

signal in corresponding yellow arrowhead-pointed cells. Scale bar in (**A**), (**E**), (**I**), (**M**), and (**Q**) represents 100 µm and scale bar in other images 50 µm. (**U–V**) Quantification of the cell density for GFP⁺ Lcp1⁺ microglia and GFP⁻Lcp⁺ DC-like cells in the dorsal (**U**), ventral (**V**), and whole area (**W**) of the brain for each genotype (n=3). Data in U-W are mean ± SEM. *p<0.05, **p<0.01 (Kruskal-Wallis test with Dunn's post-hoc).

The online version of this article includes the following figure supplement(s) for figure 8:

**Figure supplement 1.** Location of microglia in the brain ventricles of *irf8*-deficient fish.

a circular shape with few and short cellular processes (*Figure 8—figure supplement 1*). In this mutant, the density of GFP⁻; Lcp1⁺ DC-like cells was reduced in comparison to *wild-type* controls, in the ventral area (~50%) (*Figure 8U and W*).

As well-established regulators of zebrafish microglia, *csf1ra* or *csf1rb* deficiency had a strong effect on GFP⁺; Lcp1⁺ cells, with densities decreased by ~50% in the dorsal and ventral areas. Interestingly, the density of GFP⁻; Lcp1⁺ DC-like cells was reduced in the ventral part of *csf1rb* homozygous fish (~50%), while it was doubled in *csf1ra*⁻/⁻ mutant animals in comparison to *wild-type* siblings (*Figure 8I–P, U and W*). Finally, we also examined fish lacking both *csf1r* paralogs (*csf1r^{DM}*). These fish displayed a more severe phenotype, being mostly devoid of both populations of microglia and DC-like cells, as indicated by the absence of GFP and Lcp1 signal (*Figure 8Q–W*). This is consistent with previous reports that *mpeg1:GFP⁺* cells are depleted in the brain of *csf1r^{DM}* fish (*Ferrero et al., 2021*; *Oosterhof et al., 2018*).

Collectively, these results demonstrate the different mononuclear phagocyte-deficient zebrafish mutant lines have reduced numbers of microglia and exhibit distinct DC-like cell phenotypes. Our data also reveal that DC-like cells develop in an *irf8*-dependent manner and identify possible opposite functions for the *csf1r* paralogs in the maintenance of this population.

## Discussion

In the present study, we have characterized the immune microenvironment of the adult zebrafish brain by profiling total cd45⁺ leukocytes, isolated from transgenic reporter fish by FACS. First, we show that, like in mammals, microglia constitute the predominant parenchymal immune cell in the brain of the adult zebrafish. Zebrafish microglia are identified based on several common canonical markers, some of which are previously reported to be conserved in mammals (*Mazzolini et al., 2020*; *Silva et al., 2021*; *Oosterhof et al., 2018*). These include *apoeb, apoc1, lgals3bpb, ccl34b.1,* and *p2ry12*. Notably, we used different combinations of fluorescent reporter lines for the prospective isolation of adult microglia and found these genes to be consistently expressed. In addition, our observations also support a phenotypical heterogeneity of adult zebrafish microglia in the steady state by identifying several clusters sharing this microglia core signature, with different expression levels. This is in line with recent advances in our understanding of microglia diversity in human and mouse, and which revealed the presence of molecularly distinct microglia subtypes across developmental stages, specific brain regions, or disease conditions (*Stratoulias et al., 2019*; *Masuda et al., 2020*).

Although the notion of microglia heterogeneity in zebrafish is already proposed (*Silva et al., 2021*; *Wu et al., 2020*), a major finding of our study is that, surprisingly, not all parenchymal mononuclear phagocytes qualify as microglial cells. Here, we provide evidence that a proportion of myeloid cells in the healthy brain parenchyma is phenotypically distinct from microglia and identify as the zebrafish counterpart of mammalian cDC1. These cells, despite sharing the microglial expression of *mpeg1.1* and genes involved in antigen presentation, display a unique transcriptomic profile characterized by a core gene signature resembling that of mammalian cDC1 (*flt3⁺, irf8^{high}, batf3⁺, id2⁺, xcr1⁺*) but lacking canonical microglia markers. The lineage identity of these cells (referred to as DC-like cells) is further supported by their dependency to *batf3*, a key transcription factor for cDC1 development in mammals. In contrast, zebrafish microglia develop normally in the absence of *batf3*, which highlights the reliance of both populations on distinct developmental programs. This notion is also reinforced by demonstrating that, unlike microglia, zebrafish brain DC-like cells are *csf1ra*-independant. However, both populations are controlled by *irf8*, a well-established regulator of microglia differentiation and DC development in mammals (*Van Hove et al., 2019*; *Cabeza-Cabrerizo et al., 2021*).

Previously, two independent studies have reported the existence of an immune cell population with a similar expression profile to DC-like cells in the juvenile and adult zebrafish brain (*Wu et al.,*

*2020*; *Silva et al., 2021*). However, contradictory conclusions were drawn regarding the identity of these cells. In one study using bulk RNAseq, a cell population expressing *id2a*, *ccl19a.1*, *siglec15l*, but not *apoeb* or *lgals3bpb*, was identified and categorized as a phenotypically distinct microglia subtype (*Wu et al., 2020*). This population could be discriminated from other *mpeg1*-expressing parenchymal cells, notably by the lack of Tg(*ccl34b.1:GFP*) transgene expression. Interestingly, while *ccl34b.1⁺; mpeg1⁺* microglia were widely spread across brain regions, *ccl34b.1⁻ mpeg1⁺* cells showed a restricted spatial localization in the white matter. In addition, these cells also displayed a highly ramified morphology as well as independency of *csf1ra* signaling, all reminiscing the DC-like cells identified in our study. However, in another report using single-cell RNAseq, a comparable myeloid population expressing high levels of *mpeg1* as well as *ccl19a.1*, *flt3*, *siglec15l*, among other DC-like genes, was labeled as brain macrophages, owing to the absence of microglial-specific markers such as *p2ry12*, *csf1ra*, *hexb,* and *slc7a7* (*Silva et al., 2021*). The present work resolves these apparent contradictions, and provides new insights into the identity of this cell population. We report here that *ccl34b.1⁻; mpeg1⁺* cells display a similar gene signature to the DC-like cells identified in our analyses. This strongly suggests that the *ccl34b.1⁻; mpeg1⁺* and *p2ry12⁻; mpeg1⁺* populations share a similar cellular identity. Likewise, the anatomical location of the brain macrophage cluster identified by Silva and colleagues was not investigated, but based on their dominant expression profile by key DC markers, these cells likely represent the equivalent of the *p2ry12⁻; mpeg1⁺* cell population. Thus, based on the evidence that these three populations constitute a unique cell type, and coupled to the demonstration that in vivo p2ry12⁻; mpeg1⁺ cells are reliant on *batf3*, collectively these features imply these cells share more similarities with mammalian DCs than with microglia or macrophages. Therefore, based on their distinct morphology, transcriptomic signature, and *batf3*-dependency, we propose that this population represents DC-like cells rather than microglia or macrophages. Importantly, since the submission of our manuscript, the Wen lab published an independent study in which they now reclassify the ccl34b.1⁻ mpeg1⁺ cells in the zebrafish brain as cDCs, thus revising their earlier interpretation of these cells as microglia (*Zhou et al., 2023*).

One important question raised from these new findings could relate to the abundance of DC-like cells within the healthy zebrafish brain parenchyma, which is strongly different than what is known in mammals. Indeed, while murine DCs are naturally found at the brain border regions, such as the meningeal layers and the choroid plexus (structures in contact with the brain microenvironment) (*Van Hove et al., 2019*), their presence within the healthy brain parenchyma is scarce and somewhat controversial. In mammals, infiltration of functional DCs in the brain parenchyma occurs with age (*Kaunzner et al., 2012*), or following an injury or infection, where they act as important inducers of the immune response through activation of primary T cells and cytokine production (*Ludewig et al., 2016*). In addition, DC infiltration is a hallmark of several neurological diseases and aging and is believed to contribute to the establishment of a chronic neuroinflammatory state (*Ludewig et al., 2016*). In this regard, Wu et al. previously reported that *ccl34b.1⁻ mpeg1⁺* cells - or DC-like cells - exhibit functional differences, including limited mobility and phagocytic properties, and enhanced release of immune regulators following bacterial infection, when compared to *ccl34b.1⁺ mpeg1⁺* microglia. The same study also proposed that zebrafish *ccl34b.1⁻; mpeg1⁺* cells might play a regulatory role by recruiting T lymphocytes in the brain parenchyma upon infection (*Wu et al., 2020*). These biological features suggest that brain DC-like cells might exhibit APC functions. However, due to a lack of tools, this hypothesis is currently difficult to address. There is evidence that DC functionalities are conserved in teleosts (*Lugo-Villarino et al., 2010*; *Bassity and Clark, 2012*), but the process of antigen presentation in zebrafish remains poorly understood (*Lewis et al., 2014*). Because zebrafish lack apparent lymph nodes and the secondary lymphoid structures found in mammals, it is not known where stimulation of naive T cells takes place and whether fish have developed unique ways to mount an adaptive immune response. Therefore, although a comprehensive analysis of the anatomical zone enriched in DC-like cells requires further investigation, from an evolutionary perspective, it is tempting to speculate that the specific localization of zebrafish DC-like cells in the ventral brain tissue might provide an environment to facilitate antigen detection and/or presentation in this organ. Future work using the mutants as described in this study, in addition to new DC-like-specific reporter lines, will help addressing such exciting questions.

Furthermore, our work sheds light on the myeloid brain phenotype of mutant lines commonly used by the fish macrophage/microglia community. CSF1R is a master regulator of macrophage

development and function in vertebrates which is found in two copies (*csf1ra* and *csf1rb*) in zebrafish due to an extra genome duplication. Others and we have contributed to the uncovering of the relative contribution of each paralog to the ontogeny of zebrafish mononuclear phagocytes (*Herbomel et al., 2001*; *Ferrero et al., 2021*; *Hason et al., 2022*; *Oosterhof et al., 2018*). Here, we also provide a new level of precision regarding these processes. As reported, the density of all parenchymal *mpeg1:GFP⁺* mononuclear phagocytes is reduced in the brain of single *csf1ra⁻/⁻* and *csf1rb⁻/⁻* adult mutant fish, and these cells disappear when both genes are knocked out (*Oosterhof et al., 2018*; *Ferrero et al., 2021*). Using in vivo lineage tracing, we previously demonstrated that zebrafish microglia are established in two successive steps, with a definitive wave of hematopoietic stem cell (HSC)-derived adult microglia replacing an embryonic/primitive population. In addition, we showed that in *csf1rb⁻/⁻* fish remaining *mpeg1:GFP⁺* cells are of primitive origin, whereas in *csf1ra⁻/⁻* fish, they are of definitive origin (*Ferrero et al., 2021*). Collectively, these observations have led to a model in which embryonic-derived microglia make up the majority of remaining *mpeg1*-expressing cells in *csf1rb⁻/⁻* fish, while residual cells represent adult microglia in the *csf1ra⁻/⁻* line, but at a strongly reduced cell number relative to controls. However, adult microglia in these experiments were identified based on the concomitant *mpeg1:GFP* transgene expression and the HSC lineage tracing marker, a strategy that, retrospectively, did not allow to discriminate them from the DC-like cells described in this study. Here, we sought to test these models in light of our current findings, and especially following the observation that individual mutant fish exhibit opposite brain DC-like phenotypes, with DC-like cell numbers being strongly increased or decreased in *csf1ra⁻/⁻* and *csf1rb⁻/⁻* animals, respectively. In mammals, DCs arise from HSCs in the bone marrow. Although the developmental origin of zebrafish brain DC-like cells remains uncharacterized, their reduced numbers in the *csf1rb* mutant, despite their lack of *csf1rb* expression, likely reflects impaired HSC-derived definitive myelopoiesis, in line with the current model in which *csf1rb* acts at the progenitor level in the WKM to promote myeloid lineage commitment (*Ferrero et al., 2021*; *Hason et al., 2022*). Accordingly, the *csf1rb⁻/⁻* line is devoid of both populations of adult microglia and DC-like cells and, as initially proposed, the most residual cells within the brain parenchyma represent remnants of embryonic microglia (*Ferrero et al., 2021*). Conversely, the increased density of DC-like cells in *csf1ra⁻/⁻* adult fish indicates that this paralogue is dispensable for the ontogeny of DC-like cells, but points to a possible role (likely indirect) in controlling the DC-like cell growth and/or survival. This is in contrast with microglia, which we have now found to be unambiguously depleted following a *csf1ra* loss-of-function. Therefore, these findings warrant an adjustment of the initial model, as the majority of remaining *mpeg1*-expressing cells in the *csf1ra⁻/⁻* line correspond to DC-like cells, and not adult microglia. Notably, these results are consistent with the reported loss of *ccl34b.1⁺*; *mpeg1⁺* cells in *csf1ra⁻/⁻* fish by Wu and colleagues (*Wu et al., 2020*), and with the observed upregulation of DC-like genes coupled to a downregulation of microglia markers in *mpeg1*:GFP cells isolated from the brain of *csf1ra⁻/⁻ csf1rb⁺/⁻* mutant animals (*Oosterhof et al., 2018*).

In zebrafish, little is known regarding lymphoid cells in the adult CNS. Similar to DCs, lymphocytes are present in limited numbers in the healthy mammalian brain and mainly restricted to the meningeal layers, choroid plexus, or the perivascular space (*Mundt et al., 2019*; *Croese et al., 2021*). In our transcriptomic analysis, we identified an heterogeneous repertoire of lymphoid cells: T, NK, and ILCs. B lymphocytes, which could not be captured using the *cd45:DsRed* transgene (*Wittamer et al., 2011*), were also detected using the *IgM*:GFP line, albeit in very low numbers. Our data suggest that, similar to mammals, in zebrafish, lymphoid cells in the steady state are only occasionally found in the brain parenchyma, and are most likely localized in the brain border regions. Here, it is worth noting that our protocol for brain dissection requires the removal of the skull, which may completely or partially disrupt the thin meningeal layers. Consequently, whether non-parenchymal cells identified in this study are located in the meninges, in the choroid plexus or even in the blood circulation remains to be determined.

Although the innate counterparts of the lymphoid system (NK cells and ILCs) have been identified in different zebrafish organs (*Hernández et al., 2018*; *Silva et al., 2021*; *Tang et al., 2017*), the lack of specific fluorescent reporter lines has until now precluded a detailed characterization of these cell populations. In particular, as a recently discovered cell type in zebrafish (*Hernández et al., 2018*), the phenotypic and functional heterogeneity of ILC-like cells are still poorly understood. In this study, we found that the adult zebrafish brain contains a population that resembles the ILC2 subset in mammals. Like human and mouse ILC2s, these cells do not express *cd4-1*, the co-receptor for the T cell receptor

(TCR). However, they are positive for $T_H2$ cytokines *il13* and *il4*, and also express *gata3*, a transcription factor involved in ILC2 differentiation. Surprisingly, *lck* expression in our dataset was restricted to T lymphocytes and NK cells, whereas in humans, this gene is expressed across all ILC subsets (*Björklund et al., 2016*). A previous study in zebrafish reported populations representing all three ILC subtypes isolated from the intestine based on expression of the *lck:GFP* transgene (*Hernández et al., 2018*). That suggests a conserved *lck* expression pattern across species. However, in none of these experiments was the presence of ILCs in the *lck:GFP* negative fraction was investigated, so whether the absence of *lck* transcripts in our ILC2 dataset is due to a low detection sensitivity or a lack of expression remains an open question. Nevertheless, as we showed, the level of expression of ILC2 transcripts remain specifically unchanged in brain leukocytes in the context of T cell deficiency. This validates that ILC2 are indeed present in this organ. In line with this, innate-lymphoid-like cells differentially expressing *il4*, *il13,* and *gata3* have been recently annotated in the juvenile zebrafish brain (*Silva et al., 2021*).

To conclude, our study provides a single-cell transcriptomic dataset of different brain leukocyte populations and may serve as a reference to better characterize the immune cell complexity of the zebrafish brain in the steady state. Similar to mammalian microglia, zebrafish microglia are identified based on several common canonical markers, some of which are conserved between species, but their diversity is still poorly understood. Therefore, future investigations will benefit from mapping microglia heterogeneity across the zebrafish brain as a complementary approach to single-cell transcriptomics for studying microglia functions in health and disease. Further work will also be needed to elucidate the functions of some of the cell types identified in this study, especially DC-like cells, and to elucidate whether this population maintains locally or is continually replenished by cells from the periphery.

## Materials and methods

### Key resources table

| Reagent type (species) or resource | Designation | Source or reference | Identifiers | Additional information |
|---|---|---|---|---|
| Genetic reagent (*Danio rerio*) | Tg(mhc2dab:GFP_LT)sd67 | *Wittamer et al., 2011* | ZFIN: sd67 | |
| Genetic reagent (*Danio rerio*) | Tg(ptprc:DsRedexpress)sd3 | *Wittamer et al., 2011* | ZFIN: sd3 | |
| Genetic reagent (*Danio rerio*) | Tg(mpeg1.1:eGFP)gl22 | *Ellett et al., 2011* | ZFIN: gl22 | |
| Genetic reagent (*Danio rerio*) | Tg(mpeg1.1:mCherry)gl23 | *Ellett et al., 2011* | ZFIN: gl23 | |
| Genetic reagent (*Danio rerio*) | TgBAC(p2ry12:p2ry12-GFP)hdb3 | *Sieger et al., 2012* | ZFIN: hdb3 | |
| Genetic reagent (*Danio rerio*) | Tg(lck:lck-eGFP)cz1 | *Langenau et al., 2004* | ZFIN: cz1 | |
| Genetic reagent (*Danio rerio*) | TgBAC(cd4-1:mcherry)UMC13 | *Dee et al., 2016* | ZFIN: UMC13 | |
| Genetic reagent (*Danio rerio*) | Tg(Cau.Ighv-ighm:EGFP)sd19 | *Page et al., 2013* | ZFIN: sd19 | |
| Genetic reagent (*Danio rerio*) | Tg(mpx:GFP)i113 | *Mathias et al., 2009* | ZFIN: i113 | |
| Genetic reagent (*Danio rerio*) | pantheri4e1 | *Parichy et al., 2000* | ZFIN: i4e1 | |
| Genetic reagent (*Danio rerio*) | csf1rbsa1503 | Sanger Institute Zebrafish Mutation Project | ZFIN: sa1503 | |
| Genetic reagent (*Danio rerio*) | irf8std96 | *Shiau et al., 2015* | ZFIN: std96 | |

*Continued on next page*

*Continued*

| Reagent type (species) or resource | Designation | Source or reference | Identifiers | Additional information |
|---|---|---|---|---|
| Genetic reagent (*Danio rerio*) | *rag2*$^{E450fs}$ | *Tang et al., 2014* | ZFIN: *E450fs* | |
| Genetic reagent (*Danio rerio*) | *batf3*$^{ulb31}$ | This manuscript | ZFIN: *ulb31* | |
| Antibody | Anti-GFP (chicken polyclonal) | Abcam | RRID:AB_300798 | 1:500 |
| Antibody | Anti-Lcp1 (rabbit polyclonal) | In house | | 1:1000 |
| Antibody | Anti-mCherry (mouse monoclonal) | Takara Bio | RRID:AB_2307319 | 1:500 |
| Antibody | Alexa Fluor 488-conjugated anti-chicken IgG (goat polyclonal) | Abcam | RRID:AB_2636803 | 1:500 |
| Antibody | Alexa Fluor 594-conjugated anti-rabbit IgG (donkey polyclonal) | Abcam | RRID:AB_2782993 | 1:500 |
| Antibody | Alexa Fluor 647-conjugated anti-mouse IgG (donkey polyclonal) | Abcam | RRID:AB_2890037 | 1:500 |
| Commercial assay or kit | SP6 RNA Polymerase | New England BioLabs | Cat# M0207 | |
| Commercial assay or kit | High Pure PCR Cleanup Microkit | Roche | Cat# 498395500 | |
| Commercial assay or kit | RNeasy Plus mini kit | Qiagen | Cat# 74134 | |
| Chemical compound, drug | SYTOX Red | Invitrogen | Cat# S34859 | |
| Chemical compound, drug | qScript cDNA SuperMix | Quanta Biosciences | Cat# 95048–100 | |
| Software, algorithm | Flow-Jo LLC | TreeStar | RRID:SCR_008520 | |
| Software, algorithm | Black Zen software | Zeiss, Germany | RRID:SCR_018163 | |
| Software, algorithm | Blue Zen software | Zeiss, Germany | RRID:SCR_013672 | |
| Software, algorithm | R Statistical software v. 4.0.3 | R Project for Statistical Computing | RRID:SCR_001905 | |
| Software, algorithm | GraphPad Prism 8 | GraphPad software, USA | RRID:SCR_002798 | |

## Zebrafish husbandry

Zebrafish were maintained under standard conditions, according to FELASA (*Aleström et al., 2020*) and institutional (Université Libre de Bruxelles, Brussels, Belgium; ULB) guidelines and regulations. All experimental procedures were approved by the ULB ethical committee for animal welfare (CEBEA) from the ULB (protocols 842 N and 850 N). The following lines were used: *Tg(mhc2dab:GFP$_{LT}$)$^{sd67}$* (*Wittamer et al., 2011*), *Tg(ptprc:DsRed$^{express}$)$^{sd3}$* (here referred to as *cd45:DsRed*) (*Wittamer et al., 2011*), *Tg(mpeg1.1:eGFP)$^{gl22}$* (here referred to as *mpeg1:GFP*) (*Ellett et al., 2011*), *Tg(mpeg1.1:mCherry)$^{gl23}$* (here referred to as *mpeg1:mCherry*) (*Ellett et al., 2011*), *TgBAC(p2ry12:p2ry12-GFP)$^{hdb3}$* (*Sieger et al., 2012*), *Tg(lck:lck-eGFP)$^{cz1}$* (here referred to as *lck:GFP*) (*Langenau et al., 2004*), *TgBAC(cd4-1:mcherry)$^{UMC13}$* (here referred to as *cd4-1:mCherry*) (*Dee et al., 2016*), *Tg(Cau.Ighv-ighm:EGFP)$^{sd19}$* (here referred to as *ighm:GFP*) (*Page et al., 2013*), *Tg(mpx:GFP)$^{i113}$* (*Mathias et al., 2009*). The mutant lines used were: *panther$^{j4e1}$* (here called *csf1ra$^{-/-}$*)(*Parichy et al., 2000*); *csf1rb$^{sa1503}$*, generated via ethyl-nitrosurea (ENU) mutagenesis, were obtained from the Sanger Institute Zebrafish Mutation Project and previously characterized (*Ferrero et al., 2021*), *irf8$^{std96}$* (*Shiau et al., 2015*), *rag2$^{E450fs}$* (*Tang et al., 2014*). Special care was taken to control reporter gene dosage through experiments (with all control and mutant animals used in this study known to carry similar hemizygous or homozygous doses of the GFP transgenes). The term 'adult' fish refers to animals aged between 4 months and 8 months old. For clarity, throughout the text, transgenic animals are referred to without allele designations.

## Generation of *batf3*$^{-/-}$ mutant zebrafish

The *batf3* (ENSDARG00000042577) knockout mutant line was generated using the CRISPR/Cas9 system. A single guide RNA (sgRNA) targeting the ATG start in the first exon (targeting sequence:

GAAGTGATGCTCCAGCTCTA) was identified and selected for its highest on-target activity and lowest predicted off-target score using a combination of the Sequence Scan for CRISPR software (available at http://crispr.dfci.har-vard.edu/SSC/) (*Xu et al., 2015*) and the CRISPR Scan (available at http://www.crisprscan.org/). The DNA template for the sgRNA synthesis was produced using the PCR-based short-oligo method as described (*Talbot and Amacher, 2014*). The following primers were used: Fw: 5'- GCGATTTAGGTGACACTATA-3' and Rv: 5'- AAAGCACCGACTCGGTGCCAC-3'. The resulting PCR product was purified by phenol-chloroform extraction and used for in vitro transcription using SP6 RNA polymerase (NEB, M0207). The resulting sgRNA was purified using the High Pure PCR Cleanup Microkit (Roche, 498395500). 60 pg sgRNA and 100 pg Cas9 protein (PNA Bio) were co-injected into one-cell stage *wild-type* embryos. The genotyping of both embryos and adults was performed using the following primers: *batf3* fw: 5'- ACTTGACAGTTTAAGCATGCCT-3' and *batf3* rv: 5'- GAACATACCTCGCTCTGTCG-3'. PCR amplicons were analyzed using a heteroduplex mobility assay (on a 8% polyacrylamide gel) to assess the presence of CRISPR/Cas9-induced mutations.

The *batf3*[ulb31] line carries an 8 bp deletion in exon 1. The deletion introduces a frameshift after amino acid 16 of the predicted 121-amino acid ORF, followed by eight heterologous amino acids and then three successive premature stop codons. Heterozygous F1 fish were backcrossed at least four generations with AB* *wild-types* before being crossed to *Tg(mhc2dab:GFP; cd45:DsRed)* fish, as well as *Tg(p2ry12:p2ry12-GFP; mpeg1:mCherry)* animals for phenotype assessment.

## Flow cytometry and cell sorting

Cell suspensions from adult brains were obtained as previously described (*Wittamer et al., 2011*; *Ferrero et al., 2021*; *Ferrero et al., 2018*). Briefly, adult brains dissected in 0.9 X Dulbecco's Phosphate Buffered Saline (DPBS) were triturated and treated with Liberase TM at 33 °C for 30–45 min, fully dissociated using a syringe with a 26 G needle, and washed in 2% fetal bovine serum diluted in 0.9 X DPBS. Cell suspensions were centrifuged at 290 g 4 °C 10 min and filtered through a 40 µm nylon mesh. Just before flow cytometry analysis, SYTOX Red (Invitrogen) was added to the samples at a final concentration of 5 nM to exclude non-viable cells. Flow cytometry acquisition and cell sorting was performed on a FACS ARIA II (Becton Dickinson). To perform the qPCR experiments, between 7000–10,000 *cd45:DsRed* + leukocytes and approximately 2500 *p2ry12:GFP*[+]; *mpeg1:mCherry*[+] microglia or *p2ry12:GFP*[-]; *mpeg1:mCherry* [+]DC like cells were sorted, collected in RLT Plus buffer (Qiagen) and flash frozen in liquid nitrogen. Analyses were performed using FlowJo software. For morphological evaluation, 100,000 *cd45*:DsRed[+] sorted cells were concentrated by cytocentrifugation at 300 g for 10 min onto glass slides using a Cellspin (Tharmac). Slides were air-dried, fixed with methanol for 5 min, and stained with May-Grünwald solution (Sigma) for 10 min. Then, slides were stained with a 1:5 dilution of Giemsa solution (Sigma) in distilled water (dH$_2$O) for 20 min, rinsed in dH$_2$O, dehydrated through ethanol series and mounted with DPX (Sigma).

## Bulk RNA sequencing and data analysis

### Sample processing and cDNA

Cell sorting and RNA sequencing was performed as previously described (*Kuil et al., 2020*). Approximately 8000 microglial cells (*p2ry12*[+]; *cd45*[+]or *mhc2dab*[+]; *cd45*[low], n=2 for each Tg) and 1200 DC-like cells (*mhc2dab*[+]; *cd45*[high], n=2) were sorted. RNA was isolated using the miRNeasy Micro Kit (Qiagen) according to the manufacturer's instructions. RNA concentration and quality were evaluated using a Bioanalyzer 2100 (Agilent Technologies). The Ovation Solo RNA-Seq System (NuGen-TECAN) with the SoLo Custom AnyDeplete Probe Mix (Zebrafish probe set) were used to obtain indexed cDNA libraries following the manufacturer recommendation.

### Sequencing

Sequencing libraries were loaded on a NovaSeq 6000 (Illumina) using a S2 flow cell and reads/fragments were sequenced using a 200 Cycle Kit.

### Alignment and feature counting

Sequenced reads were then trimmed using *cutadapt* with default parameters except for '−−overlap 5 −−cut 5 −−minimum-length 25:25 -e 0.05.' Trimmed FQ files were at this point processed with the same approach for both datasets, including *Wu et al., 2020* expression data that were retrieved from

GEO data repository (GEO Accession: GSM4725741) (*Wu et al., 2020*). Trimmed and filtered reads were then mapped against the reference genome GRCz11.95 using STAR aligner with the '–twopass-Mode basic' and '–sjdbOverhang 100.' BAM files were then indexed and filtered using SAMTOOLS 'view -b -f 3 F 256.' Finally, transcript feature annotations for Ensembl genes using *Danio_rerio* v. GRCz11.95 were quantified using HTSeq-counts call with default parameters specifying '-r pos -s yes -a 10 −−additional-attr=gene_name -m intersection-nonempty −−secondary-alignments=ignore −−supplementary-alignments=ignore.' General sequencing and mapping stats were calculated using FastQC and MultiQC.

### Feature count matrix preprocessing, normalization, and differential expression

Feature count matrices were further preprocessed, filtering low count genes (≥ 10#) for 2 out of 2 samples in each group (this manuscript dataset) or 3 out of 5 samples per group (*Wu et al., 2020* dataset). Overall, for this manuscript dataset, we obtained 13,663 genes expressed in both replicates, whereas *Wu et al., 2020* dataset showed 9007 genes that were expressed in at least 3 of the 5 replicates. Then DESeq2 (v. 1.30.0) for R statistical computing was used to normalize the raw counts and perform differential expression analysis focusing on protein-coding genes with de-duplicated gene names (as '_#') (*Love et al., 2014*). Statistical differential expression and downstream analyses were performed using R Statistical software v. 4.0.3.

## Single-cell RNA sequencing and data analysis

### Single-cell RNA-seq library preparation and sequencing

Adult brain single-cell suspensions were prepared as described before from adult *Tg(p2ry12:GFP; cd45:DsRed)* fish (n=3), using calcein violet to exclude dead cells (1 µM, Thermo Fisher). A total of 14,000 *cd45:DsRed*[+] cells were processed for single-cell profiling using the 10 x Genomics platform and diluted to a density of 800 cells/µl following 10 x Genomics Chromium Single-cell 3' kit (v3) instructions. Library preparation was performed according to 10 x Genomics guidelines and sequenced on an Illumina NextSeq 550. Raw sequencing data was processed using the Cell Ranger with a custom-built reference based on the zebrafish reference genome GRCz11 and gene annotation Ensembl 92 in which the EGFP and DsRed sequences were included.

### Single-cell RNA-seq data preprocessing

Single-cell raw counts were processed using Seurat (v3) (*Butler et al., 2018*; *Satija et al., 2015*). Briefly, genes with zero counts for all cells were removed, and cell filters were applied for ≥ 20% reads mapping to mitochondrial genes and nFeature >300. Additionally, mitochondrial genes '^mt-' and ribosomal genes '^rp[sl]' were masked for further downstream analysis, as well as non-coding protein genes selected with the current feature annotations of the EnsemblGene 95 from GRCz11 zebrafish genome. Overall, providing a dataset of 4145 cells and 18,807 genes for single-cell data analysis. Analysis using the *scDblFinder R package* found no evidence of doublet enrichment, indicating that the clusters were sufficiently robust to capture normal cell states.

### Single-cell normalization, clustering, and marker genes

Single-cell filtered data were normalized using Seurat's SCTransform method with the following custom parameters: 'variable.features.n=4000 and return.only.var.genes=F.' Then, the nearest neighbour graph was built with 40 PCA dimensions, and clusters were identified using manually selected resolution based on the supervised inspection of known markers leading to the optimal 'resolution = 0.6 (Louvain)' and 'n.neighbors=20.' The same parameters were used for the dimensionality reduction as UMAP. Finally, cluster annotation was performed by inspecting the identified marker genes using the FindAllMarkers function (one v. rest with default parameters except for 'min.pct=0.25').

### Single cell pathway analysis

For pathway analysis, marker genes of all four microglia clusters together (MG1, MG2, MG3, MG4) or all four DC-like clusters (DC1, DC2, DC3, DC4) were obtained using Seurat's FindMarkers function. Next, differentially expressed zebrafish genes (log2 fold-change>0.25, p-adjusted<0.05) were

converted to their human orthologs using the gProfiler tool (*Raudvere et al., 2019*) and validated using the ZFIN (https://zfin.org/) and Alliance Genome databases (https://www.alliancegenome.org/). Genes with no corresponding orthologs were not included. From this gene list, Gene Ontology terms (Biological Processes) and Reactome pathways were obtained using the Cytoscape ClueGO application (two-sided hypergeometric statistical test, Bonferroni correction) (*Bindea et al., 2009*). To explore MG and DC-like conserved cell type signatures (*Supplementary file 4*), each gene list was uploaded to the Enrichr database (*Xie et al., 2021*) to identify the most enriched 'Cell Types' categories, querying PanglaoDB (*Franzén et al., 2019*).

## Quantitative PCR

RNA extraction was performed using the RNeasy Plus Mini kit (Qiagen) and cDNAs were synthesized using the qscript cDNA supermix (Quanta Biosciences), as previously described (*Ferrero et al., 2018*). Reactions were run on a Bio-Rad CFX96 real-time system (Bio-Rad), using the Kapa SYBR Fast qPCR Master Mix (2 X) kit (Kapa Biosystems) under the following thermal cycling conditions: 3 min at 95 °C and 40 cycles of 5 s at 95 °C, 30 s at 60 °C. A final dissociation at 95 °C for 10 s and a melting curve from 65 to 95°C (0.5 °C increase every 5 s) were included to verify the specificity and absence of primer dimers. Biological replicates were compared for each subset. Relative amount of each transcript was quantified via the ΔCt method, using *elongation factor 1 alpha* (*eef1a1l1*; ENSDARG00000020850) expression for normalization.

## Immunostaining and vibratome sections

Adult brains were dissected, fixed in 4% PFA, and incubated overnight in 30% sucrose in PBS before being snap-frozen in OCT (Tissue-Tek, Leica) and stored at –80 °C. Immunostaining was performed on 14 μm cryosections as previously described (*Ferrero et al., 2018*). The following primary and secondary antibodies were used: chicken anti-GFP polyclonal antibody (1:500; Abcam, Cat# ab13970), rabbit anti-Lcp1 (1:1000), mouse anti-mCherry monoclonal antibody (1:500; Takara Bio Cat# 632543), Alexa Fluor 488-conjugated anti-chicken IgG antibody (1:500; Abcam Cat# ab150169), Alexa Fluor 594-conjugated anti-rabbit IgG (1:500; Abcam Cat# ab150076), Alexa Fluor 647-conjugated anti-mouse IgG (1:500; Abcam Cat# ab150107). For vibratome sections, adult brains were fixed in 4% PFA, embedded in 7% low-melting agarose in PBS, and sectioned at 100 μm using a vibratome (Leica). Sections were mounted directly with Glycergel (Dako) and imaged without prior immunostaining to visualize endogenous *cd45:DsRed* fluorescence.

## Imaging and image analyses

Fluorescent samples were imaged using a Zeiss LSM 780 inverted microscope (Zeiss, Oberkochen, Germany), with a Plan Apochromat 20 X objective. Image post-processing was performed using Zeiss Zen Software (ZEN Digital Imaging for Light Microscopy), as previously described (*Ferrero et al., 2021*). Cells were manually counted using the Black Zen software and divided by the area of the brain section (cell density/μm$^2$) and quantified between 5–11 transversal sections per brain. Cytospuned cells were imaged using a Leica DM 2000 microscope equipped with a 100 X objective and scanned using a NanoZoomer-SQ Digital Slide scanner (Hamamatsu).

## Data collection

The sample size was chosen based on previous experience in the laboratory, for each experiment to yield high power to detect specific effects. No statistical methods were used to predetermine sample size and experiments were repeated at least twice. Homozygous mutant animals used in this study were obtained by heterozygous mating. No fish were excluded. Genotyping was performed on tail biopsies collected from individual euthanized fish, in parallel to brain dissection. Randomly selected samples for each genotype were then immunostained in one batch, assessed phenotypically in a blind manner, and grouped based on their genotype.

## Statistical analyses

Statistical differences between mean values of two experimental groups were analyzed by Student's t-test or the equivalent U-Mann-Whitney non-parametric test, when parametric assumptions were not met in the sample data. Results are expressed as mean ± standard of the mean (SEM) and considered

to be significant at $p<0.05$. Details on the number of fish (biological replicates) used in each experiment, the statistical test used, and statistical significance are indicated in each figure and figure legends. Statistical analyses were performed using GraphPad Prism 8.

## Acknowledgements

We thank all members of the Wittamer lab and Sumeet Pal Singh for critical discussion and comments on the manuscript. We are also grateful to Marianne Caron for technical assistance and to Daniel M Borràs for guidance with bioinformatic analyses. We also acknowledge Christine Dubois for support with flow cytometry, F Libert and A Lefort from the ULB Genomic Core Facility, and S Reinhardt, A Kränkel and A Petzold at the Dresden-Concept Genome Center in Germany. This work was funded in part by the Funds for Scientific Research (FNRS) under Grant Numbers F451218F, UN06119F, and UG03019F, the program ARC from the Wallonia-Brussels Federation, the Alzheimer Research Foundation (SAO-FRA), the Minerve Foundation (to VW), the Fonds David et Alice Van Buuren, the Fondation Jaumotte-Demoulin, and the Fondation Héger-Masson (to VW, GF, and MM). MR is supported by a Chargé de Recherche fellowship (FNRS), GF, and AM by a Research Fellowship (FNRS) and MM by a fellowship from The Belgian Kid's Fund.

## Additional information

### Funding

| Funder | Grant reference number | Author |
|---|---|---|
| Fonds de la Recherche Scientifique | F451218F | Valerie Wittamer |
| Fonds de la Recherche Scientifique | UN06119F | Valerie Wittamer |
| Fonds de la Recherche Scientifique | UG03019F | Valerie Wittamer |
| program ARC from the Wallonia-Brussels Federation | | Valerie Wittamer |
| Minerve Foundation | | Valerie Wittamer |
| Alzheimer Research Foundation (SAO-FRA) | | Valerie Wittamer |
| Fonds David et Alice Van Buuren | | Giuliano Ferrero |
| The Belgian Kid's Fund | | Magali Miserocchi |
| Chargé de Recherche fellowship (FNRS) | | Mireia Rovira |

The funders had no role in study design, data collection and interpretation, or the decision to submit the work for publication.

### Author contributions

Mireia Rovira, Conceptualization, Formal analysis, Supervision, Investigation, Visualization, Methodology, Writing – original draft, Project administration, Writing – review and editing; Giuliano Ferrero, Conceptualization, Formal analysis, Investigation, Visualization, Methodology, Writing – original draft; Magali Miserocchi, Alice Montanari, Investigation, Methodology; Ruben Lattuca, Software; Valerie Wittamer, Conceptualization, Supervision, Funding acquisition, Validation, Investigation, Visualization, Methodology, Writing – original draft, Writing – review and editing

### Author ORCIDs

Mireia Rovira ⬡ https://orcid.org/0000-0003-4050-1662
Giuliano Ferrero ⬡ https://orcid.org/0000-0001-5382-9873

Magali Miserocchi ⓘ https://orcid.org/0000-0002-4120-5833
Alice Montanari ⓘ https://orcid.org/0000-0003-2863-4761
Ruben Lattuca ⓘ https://orcid.org/0009-0001-8323-3170
Valerie Wittamer ⓘ https://orcid.org/0000-0003-0003-2646

### Ethics

This study was performed in strict accordance with the Federation of European Laboratory Animal Science Associations (FELASA) and institutional (Université Libre de Bruxelles, Brussels, Belgium; ULB) guidelines and regulations. All experimental procedures were approved by the ULB ethical committee for animal welfare (CEBEA) from the ULB (protocols 842N and 850N).

Reviewer #1 (Public review): https://doi.org/10.7554/eLife.91427.3.sa1
Reviewer #2 (Public review): https://doi.org/10.7554/eLife.91427.3.sa2
Reviewer #3 (Public review): https://doi.org/10.7554/eLife.91427.3.sa3
Author response https://doi.org/10.7554/eLife.91427.3.sa4

## Additional files

### Supplementary files

Supplementary file 1. List of cluster marker genes for identified cell types.

Supplementary file 2. Top 50 markers for each identified cell type.

Supplementary file 3. Differentially expressed genes between mononuclear phagocyte clusters.

Supplementary file 4. .Pathway enrichment analysis for mononuclear phagocyte cluster markers.

Supplementary file 5. Differentially expressed genes between dendritic cell (DC)-like versus microglia clusters (bulk RNA seq analyses).

MDAR checklist

### Data availability

All datasets and material generated for this study are included in the manuscript and Supplementary Files. Raw data for single cell RNA-seq samples and RNA-seq are available in the ArrayExpress database as accession number E-MTAB-13223 and E-MTAB-13228, respectively. The transcriptomic atlas generated in this study is available as a searchable database at: https://scrna-analysis-zebrafish.shinyapps.io/scatlas/. The code for the Shiny app is deposited at https://github.com/rulatt/scAtlas_Zebrafish/ (copy archived at *Lattuca, 2025*).

The following datasets were generated:

| Author(s) | Year | Dataset title | Dataset URL | Database and Identifier |
|---|---|---|---|---|
| Rovira M, Ferrero G, Miserocchi M, Montanari A, Lattuca R, Wittamer V | 2025 | A single-cell transcriptomic atlas reveals a new myeloid parenchymal population in the zebrafish brain | https://www.ebi.ac.uk/biostudies/ArrayExpress/studies/E-MTAB-13223?query=E-MTAB-13223 | ArrayExpress, E-MTAB-13223 |
| Rovira M, Ferrero G, Miserocchi M, Montanari A, Lattuca R, Wittamer V | 2025 | Transcriptomic analysis of microglia and DC-like cells sorted using different reporter lines | https://www.ebi.ac.uk/biostudies/ArrayExpress/studies/E-MTAB-13228?query=E-MTAB-13228 | ArrayExpress, E-MTAB-13228 |

The following previously published dataset was used:

| Author(s) | Year | Dataset title | Dataset URL | Database and Identifier |
|---|---|---|---|---|
| Wu S, Nguyen LTM, Pan H, Hassan S, Dai Y, Xu J, Wen Z | 2020 | Whole-transcriptome RNA-seq analysis of zebrafish microglia subpopulations | https://www.ncbi.nlm.nih.gov/geo/query/acc.cgi?acc=GSE156158 | NCBI Gene Expression Omnibus, GSE156158 |

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
