## [Editor Report · eLife Assessment]

This **important** work represents an advance in our understanding of resident myeloid cells in the zebrafish brain, particularly as it provides a molecular definition of dendritic cell subtypes associated with their localization. Combined evidence from single cell transcriptomics and histology is **compelling**. The associated atlas will be used as a resource by the zebrafish community and beyond.

---

## [Referee Report · Reviewer #1 (Public review)]

Using several zebrafish reporter lines, the authors characterized immune cells in the adult zebrafish brain, identifying a population of DC-like cells with distinct regional distribution and transcriptional profiles. These cells were distinct from microglia and other macrophages, closely resembling murine cDC1s. Analysis of different mutants revealed that this DC population depends on Irf8, Batf3 and Csf1rb, but not Csf1ra.

This elegantly designed study provides compelling evidence for additional heterogeneity among brain mononuclear phagocytes in zebrafish, encompassing microglia, macrophages, and DC-like cells. It advances our understanding of the immune landscape in the zebrafish brain and facilitates better distinction of these cell types from microglia.

---

## [Referee Report · Reviewer #2 (Public review)]

The authors made an atlas of single-cell transcriptome of on a pure population of leukocytes isolated from the brain of adult Tg(cd45:DsRed) transgenic animals by flow cytometry. Seven major leukocyte populations were identified, comprising microglia, macrophages, dendritic-like cells, T cells, natural killer cells, innate lymphoid-like cells and neutrophils. Each cluster was analyzed to characterize subclusters. Among lymphocytes, in addition to 2 subclusters expressing typical T cell markers, a group of il4+ il13+ gata3+ cells was identified as possible ILC2. This hypothesis is supported by the presence of this population in rag2KO fish, in which the frequency of lck and zap70+ cells is strongly reduced. The use of KO lines for such validations is a strength of this work (and the zebrafish model).

The subcluster analysis of mpeg1.1 + myeloid cells identified 4 groups of microglial cells, one novel group of macrophage like cells (expressing s100a10b, sftpbb, icn, fthl27, anxa5b, f13a1b and spi1b), and several groups of DC like cells expressing the markers siglec15l, ccl19a.1, ccr7, id2a, xcr1a.1, batf3, flt3, chl1a and hepacam2.Combining these new markers and transgenic reporter fish lines, the authors then clarified the location of leukocyte subsets within the brain, showing for example that DC-like cells stand as a parenchymal population along with microglia. Reporter lines were also used to perform detailed analysis of cell subsets, and cross with a batf3 mutant demonstrated that DC like cells are batf3 dependent, which was similar to mouse and human cDC1. Finally, analysis of classical mononuclear phagocyte deficient zebrafish lines showed they have reduced numbers of microglia but exhibit distinct DC-like cell phenotypes. A weakness of this study is that it is mainly based on FACS sorting, which might modify the proportion of different subtypes.

This atlas of zebrafish brain leukocytes is an important new resource to scientists using the zebrafish models for neurology, immunology and infectiology, and for those interested in the evolution of brain and immune system.

---

## [Referee Report · Reviewer #3 (Public review)]

Rovira, et al., aim to characterize immune cells in the brain parenchyma and identify a novel macrophage population referred to as "dendritic-like cells". They use a combination of single-cell transcriptomics, immunohistochemistry, and genetic mutants to conclude the presence of this "dendritic-like cell" population in the brain. The strength of this manuscript is the identification of dendritic cells in the brain, which are typically found in the meningeal layers and choroid plexus. In addition, Rovira, et al., findings are supported by the findings of the Wen lab and a recent Cell Reports paper. Congratulations on the nice work!

---

## [Author Response]

The following is the authors’ response to the original reviews.

**Reviewer #1 (Public Review):**
Weaknesses:While scRNA-seq data clearly revealed different subsets of microglia, macrophages, and DCs in the brain, it remains somewhat challenging to distinguish DC-like cells from P2ry12- macrophages by immunohistochemistry or flow cytometry.

Indeed, in flow cytometry analyses of adult brain samples, the p2ry12^-^; mpeg1^+^ fraction could, in theory, encompass not only DC-like cells but also other macrophage subsets, as well as B cells, since B cells have been reported to express mpeg1 in zebrafish (Ferrero et al., 2020; Moyse et al., 2020). Nevertheless, our data strongly indicate that within the brain parenchyma, DC-like cells represent the predominant component of this population. This conclusion is supported by the pronounced reduction of p2ry12^-^; mpeg1^+^ cells in brain sections from ba43 mutants, in which DC development is impaired. Currently, further phenotypic resolution is constrained by the limited availability of zebrafish-specific antibodies and the restricted palette of fluorescent reporter lines capable of distinguishing MNP subsets. We anticipate that future efforts, including the generation of novel transgenic lines informed by our dataset (initiatives already underway in our group), will enable more precise discrimination among these distinct subsets.

**Reviewer #2 (Public Review):**
A weakness of this study is that it is mainly based on FACS sorting, which might modify the proportion of different subtypes.

We agree that reliance solely on FACS could potentially introduce biases in the proportions of different subtypes. To minimize this concern, we complemented our flow cytometry data with quantification performed directly on brain sections using immunohistochemistry. This approach allowed us to validate cell population distributions in situ, thereby confirming that the trends observed by FACS accurately reflect the cellular composition of microglia and DC-like cells within the brain parenchyma.

**Reviewer#3 (Public Review):**
A weakness is the lack of specific reporters or labeling of this dendritic cell population using specific genes found in their single-cell dataset. Additionally, it is difficult to remove the meningeal layers from the brain samples and thus can lead to confounding conclusions. Overall, I believe this study should be accepted contingent on sufficient labeling of this population and addressing comments.

While the generation of DC-like specific transgenic lines is indeed a promising direction (and such efforts are currently underway in our group), creating and validating these lines is time-consuming. Importantly, although these additional tools will be valuable for future functional investigations, we believe they would not impact the main conclusions or core message of our current work, where we already provide detailed spatial information on DC-like cells, and we demonstrated their lineage identity through the use of our newly generated batf3 mutant line.

**Recommendations for the authors:**
Major Comments:The authors should discuss another recent report demonstrating DCs in the zebrafish brain, which also developed independently of Csf1ra, and compare the two datasets (Zhou et al. Cell reports, 2023).

Thank you for highlighting the study by Zhou et al., which offers complimentary insight into the dendritic cell population in the zebrafish brain. We note that in this work, the authors reclassify ccl34b.1^-^ mpeg1^+^ brain-resident cells as conventional DCs, thus revising their earlier interpretation of these cells as microglia (Wu et al., 2020). This shift in interpretation is based on their transcriptional comparison between the previously characterized ccl34b.1^-^ mpeg1^+^ population and a new dataset of brain

mpeg1^+^ cells. This updated classification aligns closely with our findings. Given that our data already demonstrate the equivalence between the DC-like cells described in our study and the ccl34b.1^-^ mpeg1^+^ population, repeating a direct transcriptional comparison would be redundant. We have now included a discussion of this work in the revised manuscript. Specifically, we have added the following sentences in the discussion: “Importantly, since the submission of our manuscript, the Wen lab published an independent study in which they now reclassify the ccl34b.1^-^ mpeg1^+^ cells in the zebrafish brain as cDCs, revising their earlier interpretation of these cells as microglia (Zhou et al., 2023)”.

Data reported in Figure 5 should be quantified (cell numbers, how many brains analyzed).

Thank you for this comment. We would like to clarify that the primary purpose of Figure 5 (and Figure 5 supplement 1) is to provide an initial qualitative overview of the different MNP subsets present in the adult brain, using the currently available transgenic and immunohistochemical tools. These descriptive analyses were instrumental in identifying the most reliable combination, namely the Tg(p2ry12:p2ry12GFP; mpeg1.1:mCherry) double transgenic line in conjunction with L-plastin immunostaining, to distinguish microglia from other parenchymal MNPs. Quantitative analyses using this optimized strategy are presented in Figure 7 (Figure 7 supplement 1), where we systematically enumerate the different MNPs. We therefore believe that performing additional quantification in Figure 5 would be redundant with the more robust data already shown in Figure 7. As requested, we have now included in the Figure 5 legend that images are representative of brain tissue sections from 2-3 fish.

The title mentions an "atlas", but there is no searchable database or website associated with the paper. Please provide one.

We agree and fully support the importance of data accessibility. To facilitate use of our dataset by the scientific community, we have developed a user-friendly, searchable web interface that allows users to explore gene expression pacerns within our dataset. This website is available at https://scrna-analysis zebrafish.shinyapps.io/scatlas/

This information has now been included in the “Data availability statement” section of the manuscript.

**Reviewer #1 (Recommendations For The Authors):**
Specific comments:The authors should discuss another recent report demonstrating DCs in the zebrafish brain, which also developed independently of Csf1ra, and compare the two datasets (Zhou et al. Cell reports, 2023).

Thank you for this suggestion. Please refer to our response in the major comments section, where we address this point in detail.

Within macrophages, the authors identified 5 clusters including 4 microglia clusters and 1 MF cluster (Figure 4). Does the laUer relate to 'BAMs' and express markers previously described in murine BAMs, including Lyve1, CD206, etc.? Or to monocytes? By flow cytometry, monocytes were detected (Figure 1B), but not by scRNA-seq.

You have raised an important point here. As described in lines 197-202 (“results” section), the cells in the MF cluster exhibit a macrophage identity, based on their expression of classical macrophage markers such as marco, mfap4 or csf1ra. However, we were unable to confidently annotate this cluster more specifically. We also considered whether this population might resemble mammalian BAMs or monocytes, cell types that, to our knowledge, have not yet been clearly identified in zebrafish. However, orthologous markers typically associated with murine BAMs were not detected (lyve1) or not specifically enriched (mrc1a/mrc1b) in the MF cluster (see below). Based on these findings, we can only cautiously propose that this cluster may represent blood-derived macrophages and / or monocytes.

To further address your suggestion, we performed a cell type enrichment analysis using the marker genes of the MF cluster, following the same strategy as for the microglia and DC-like clusters presented in Figure 4 supplement 2 C,D. This analysis revealed significant for “monocytes” and “macrophages”, further supporting a general monocytic/macrophage identity (see below). At present, further characterization of this cluster is limited by the lack of zebrafish-specific antibodies and the restricted palette of fluorescent reporter lines that distinguish among MNP subsets. We anticipate that future studies, including the development of new transgenic lines guided by our dataset, will allow for a more precise analysis of this distinct population.

**Author response image 1. sa4fig1:** 

Do all 4 DC clusters identified by scRNA-seq represent cDC1s? or are there also cDC2s, and cDC3s present?

In our analyses, the four dendritic cell clusters identified by scRNA-seq (DC1-DC4) exhibit transcriptional profiles consistent with a conventional type 1 dendritic cell (cDC1) identity. These clusters uniformly express hallmark cDC1-associated genes, while lacking expression of markers typically associated with mammalian cDC2 or plasmacytoid dendritic cells (pDCs). For instance, irf4, a key transcription factor required for cDC2 development, is not detected in our dataset. Similarly, we do not observe expression of genes characteristic of pDCs.

That said, the absence of cDC2 or pDC-like signatures in our dataset does not rule out the presence of these populations in zebrafish.

While they show that DC-like cells did not express Csf1rb (Figure 4D) or other macrophage/microglia genes, DC-like cells were affected in the Csf1rb mutants and in double mutants, demonstrating that their development depends on Csf1rb signaling, as known for macrophages but not DCs. Can the authors discuss this in more detail with regard to DC differentiation/precursors?

Thank you for pointing this out. As previously demonstrated, CSF1R signaling in zebrafish is more complex than in mammals, due to the presence of two paralogs, csf1ra and csf1rb, which exhibit partially non-overlapping functions (Ferrero et al., 2021). We and others have shown that csf1rb signaling is implicated in the regulation of definitive hematopoiesis, particularly in the regulation of hematopoietic stem cell (HSC)-derived myelopoiesis. Although the developmental origin of zebrafish brain DC-like cells remains uncharacterized, their reduced numbers in the csf1rb mutant, despite their lack of csf1rb expression, supports the current model in which csf1rb acts at the progenitor level, promoting myeloid lineage commitment. According to this, csf1rb disruption affects the differentiation of multiple myeloid subsets, which likely include DC-like cells. We have developed this point in the discussion section (lines 502506).

Do the DCs express Csf1ra?

Csf1ra transcripts are not found in DCs in our dataset. As shown below, csf1ra expression is restricted to the microglia and macrophage clusters. These observations are in line with those made by Zhou et al., 2023.

**Author response image 2. sa4fig2:** 

Fig. 5, the number of brains analyzed should be added, and also quantifications of cell numbers included. It is mentioned (line 260) that P2ry12GFP+mpeg1mCherry+ microglia are abundant across brain regions while P2ry12GFP- mpeg1mCherry+ cells particularly localize in the ventral part of the posterior brain parenchyma. It would be nice if images of the different brain regions were provided.

Regarding the quantification, we refer to our response in the major comments section, where we explain that detailed quantification of microglia and other MNP subsets is provided in Figure 7, using a more refined strategy for distinguishing cell types.

As requested, we have now included representative sections from the forebrain, midbrain and hindbrain of adult Tg(mhc2dab:GFP; cd45:DsRed) fish. These images illustrate the spatial distribution of DC-like cells across brain regions. Notably, DC-like cells are most abundant in the ventral areas of the midbrain and hindbrain, and are also present in the posterior telencephalon, particularly concentrated in the region of the commissura anterior. This regional annotation is based on the zebrafish brain atlas by Wullimann et al., 1996 (Neuroanatomy of the zebrafish brain, https://doi.org/10.1007/978-3-0348-8979-7).

These additional images have been included in Figure 5 Supplement 1 (A-E).

It is sometimes not evident whether the Pr2y12- cells included DC-like cells and macrophages, which should be discussed.

Thank you for bringing this to our attention. Upon review, we agree this point required clearer explanation throughout the text, particularly beginning with the description of putative DC-like cells in Figure 5. We have now revised the manuscript to improve clarity and becer guide readers through the phenotypic identification of DC-like cells using the Tg(p2ry12:p2ry12-GFP;mpeg1:mCherry) line. Specifically, we have modified the titles in the results section from page 5 to page 9, so that readers can more easily follow the step-by-step approach we used to distinguish DC-like cells from microglia.

To directly address your comment: the p2ry12^-^; mpeg1^+^ fraction may, in theory, include not only DC-like cells but also other macrophage subsets and B cells, as B cells have been shown to express mpeg1 in zebrafish (Ferrero et al., 2020; Moyse et al., 2020). Nevertheless, our data strongly indicate that within the brain parenchyma, DC-like cells represent the predominant component of this population. This conclusion is supported by the pronounced reduction of p2ry12^-^; mpeg1^+^ cells in brain sections from ba43 mutants, in which DC development is impaired.

We have revised the text accordingly to clarify this point in the results section of the manuscript (line 355).

For example, the DC-like cell population in Figure 6C appears to include two populations of cells. Thus, it is unclear whether the sorted mhc2dab:GFP+;CD45:DsRedhi population for bulk-seq also contains the MF population identified in Fig. 2.

Thank you for this thoughtful observation. During the course of this study, we indeed considered how best to isolate non-microglial macrophages in order to specifically recover the MF population identified in our scRNA-seq analysis. However, with the current repertoire of fluorescent transgenic zebrafish lines, it remains technically challenging to selectively isolate non-microglial macrophages from the adult brain. As a result, the mhc2dab:GFP_+_; cd45:DsRedhi sorted population used for bulk RNA-seq may indeed include a mixture of DC-like and other mononuclear phagocytes, potentially the MF population. In contrast, our data demonstrate that the Tg(p2ry12:p2ry12-GFP) line provides a more selective tool for isolating microglia, minimizing contamination from other mononuclear phagocyte subsets.

In Figure 7, a reduction of GFP-mpeg+ cells can be seen in baf3 mutants. Could the remaining cells be the (non-microglia) macrophages? Or in Figure 8, could the remaining P2ry12GFP-Lcp1+ cells in Irf8 mutants be macrophages?

Indeed, we believe it is likely that the remaining mpeg1^+^ cells observed in ba43 mutants include non-microglial macrophages and/or B cells, as we and others previously showed that zebrafish B cells express mpeg1.1 transcripts and are labeled in the mpeg1.1 reporters (Ferrero et al., 2020). This interpretation is further supported by the observation that the reduction in mepg1+ cells is more pronounced in brain sections than in flow cytometry samples, where non-parenchymal mpeg+ cells, such as peripheral macrophages or B cells, are likely enriched. To explore this possibility, we attempted to assess the expression of MF- and B cell-specific markers in the remaining mpeg1+ population isolated from ba43 mutants. However, due to the very low numbers of cells recovered per animal, we were limited to analyzing only a few markers. Despite multiple attempts, qPCR analyses proved unconclusive, likely due to low transcript abundance. We thank you for your understanding of the technical limitations that currently prevent a more definitive characterization of these remaining cells.

Regarding the irf8 mutants (Figure 8), irf8 is a well-established master regulator of mononuclear phagocyte development. In mice, deficiency results in developmental defects and functional impairments across multiple myeloid lineages, including microglia, which exhibit reduced density (Kierdorf et al., 2013) and an immature phenotype (Vanhove and al., 2019). Similarly, in zebrafish, irf8 mutants show abnormal macrophage development, with an accumulation of immature and apoptotic cells during embryonic and larval stages (Shiau et al., 2014). Based on these findings, it is plausible that the residual p2ry12:GFP^-^ Lcp1^+^ cells observed in the irf8 mutant brains represent immature or arrested mononuclear phagocytes, possibly including both microglia and DC-like cells. This is supported by their distinct morphology and specific localization along the ventricle borders. However, as previously noted, our current tools do not permit to conclusively identify these cells.

**Reviewer #2 (Recommendations For The Authors):**
A few sentences are not easy to understand for a "non zebrafish specialist".(1) Page 3 line 111 The sentence "Interestingly, analyses of brain cell suspensions from double transgenics showed p2ry12:GFP+ microglia accounted for half of cd45:DsRed+ cells (50.9 % {plus minus} 2.9; n=4) (Figure 1D,E). Considering that mpeg1:GFP+ cells comprised ~75% of all leukocytes, these results indicated that approximately 25% of brain mononuclear phagocytes do not express the microglial p2ry12:GFP+ transgene." is not clear. This point is significant and deserves a more detailed explanation.

We apologize for the lack of clarity in this section. The quantification presented in Figure 1 refers specifically to cd45:Dsred^+^ leukocytes, meaning that the reported percentages of p2ry12:GFP^+^ and mpeg1:GFP^+^ cells are calculated relative to the total cd45+ population (defined as 100%). Specifically, we observed that approximately 51% of all cd45+ cells were p2r12:GFP^+^ microglia, while around ti5% were mpeg1:GFP^+^. From these values, we infer that about 25% of mpeg1:GFP^+^ leukocytes do not express the p2ry12:GFP transgene and therefore likely represent non-microglial mononuclear phagocytes. We agree that this distinction is important and have revised the text accordingly to clarify the interpretation for readers who may be less familiar with zebrafish transgenic lines or gating strategies. See page 3, lines 107 117.

(2) Line 522; Like human and mouse ILC2s, "these cells do not express the T cell receptor cd4-1" is confusing (T cell receptor should be reserved to the ag specific TCR). Also, was TCR isotypes expression analyzed (and how was genome annotation used in this case ?)

Thank you for this insightful comment. We agree that the term “T cell receptor” should be used specifically to refer to antigen-specific TCRs, and we have revised the discussion accordingly to avoid any confusion. Regarding your question on the analysis of TCR isotype expression and the use of genome annotation: due to technical limitations, we did not pursue TCR isotype-level analysis in this study. Instead, we relied on established markers such as cd4-1 and cd8a to distinguish T cell populations, acknowledging that cd4-1 is not expressed by ILC2-like cells in our dataset. We have clarified these points in the relevant sections of the manuscript (see lines 168 and 535)

The analysis of single-cell data might be more detailed, with more explanation about possible doublet identification and normalization procedures.

Thank you for highlighting the need for additional clarity regarding our scRNA-seq analysis.

As noted in the Seurat tutorial, “cell doublets or multiplets often exhibit abnormally high gene count” (https://sa7jalab.org/seurat/archive/v3.0/pbmc3k_tutorial). To evaluate this, we performed a dedicated doublet detection analysis using the scDblFinder R package. Our results indicated that the proportion of predicted doublets is low (see Figure below), and when present, these doublets are distributed among the different clusters. This contrasts with the typical clustering of doublets into discrete groups and indicates that our single-cell sequencing workflow was sufficiently robust to predominantly capture singlets.

Regarding normalization, we have clarified this in the manuscript. Briefly, single-cell data were normalized using Seurat’s SCTransform method with the following custom parameters: “variable.features.n=4000 and return.only.var.genes=F”. These settings are now clearly described to ensure reproducibility.

**Author response image 3. sa4fig3:** 

**Reviewer #3 (Recommendations For The Authors):**
Major issuesThough baf3 mutants were generated the manuscript will greatly benefit from in situ labeling by RNAscope or the generation of transgenic reporters to conclusively localize this dendritic cell population and address any potential contamination issues.

We thank you for this constructive suggestion. We agree that in situ labeling approaches such as RNAscope would offer valuable complementary insights. In our current study, however, we already provide detailed spatial information on DC-like cells, and we demonstrated their lineage identity through the use of our newly generated batf3 mutant line.

To address concerns regarding potential contamination, we have carefully analyzed more than two dozens adult brains to date and consistently observed abundant DC-like cells within the brain parenchyma, exhibiting a reproducible and specific spatial distribution, as described in the manuscript. This consistent localization across multiple samples strongly supports the genuine presence of these cells in the brain rather than artifactual contamination.

While the generation of DC-like specific transgenic lines is indeed a promising direction (and such efforts are currently underway in our group) we note that creating and validating these lines is time-consuming and falls beyond the scope of the present study. Importantly, although these additional tools will be valuable for future functional investigations, we believe they would not impact the main conclusions or core message of our current work.

The morphological characterization of CD45:DsRed+ macrophages stained with May-Grunwald-Giemsa has been previously reported in the paper, "Characterization of the mononuclear phagocyte system in the zebrafish" Wittamer et al., 2011."Morphologic analyses revealed that the majority of cells exhibited the characteristics of monocytes/macrophages namely low nuclear to cytoplasm ratios and a high number of cytoplasmic vacuoles (Figure 3B).

We thank you for pointing out the reference to Wittamer et al., 2011. In that study, we indeed provided the first morphological characterization of mononuclear phagocytes (MNPs) in various adult zebrafish organs using the cd45:DsRed line in combination with the mhc2dab:GFP reporter. The focus was primarily on MNPs across peripheral tissues. In the current study, our aim is broader: we investigate the full diversity of brain immune cells, using cd45 as a general marker for leukocytes. As part of this comprehensive characterization, we applied MGG staining, a widely accepted cytological technique, to gain morphological insight into the sorted CD45:DsRed+ population. This method remains a valuable and rapid approach to visually assess cell type heterogeneity, especially when evaluating samples where multiple immune cell lineages may be present.

While there is some overlap with the methodology used in Wittamer et al., the context, scope, and tissue examined differ substantially. Thus, the inclusion of MGG staining in this study serves to complement our broader transcriptomic analyses by providing supporting morphological evidence specific to brain-resident immune cells.

We have now clarified this distinction in the revised manuscript to better differentiate the current work from our previous findings (see line 85).

Figure 5 data should be quantified.

Please refer to our response in the major comments section, where we address this question in detail.

Figure 7- Figure Supplement 1. J, K has no CD45:DsRed positive cells in baf3 mutants, which is counterintuitive because CD45:DsRed should capture all hematopoietic cells and is not specific to dendritic cells.

It is correct that cd45 is a general leukocyte marker, labeling all immune cells, including dendritic cells. In this Figure, we used the Tg(cd45:DsRed) transgenic line to visualize the phenotype because it offers an alternative to IHC, with the advantage of strong endogenous fluorescence and easier screening of vibratome sections. However, this technique has limitations: due to fixation, only cells with high fluorescence (e.g. cd45^high^dendritic cells) are captured, while those with medium/low expression (e.g. cd45^low^ microglia) are often not visible. This explains why fewer cells are observed in both wild-type and ba43 mutant brains (Figure 5 KN, Figure 7 – supplement 1 JK). While this approach is quicker and allows for thicker sections, IHC remains the preferred method for the rest of the analyses, including the use of additional markers to identify all relevant cell populations.

Thank you for bringing this point of confusion to our attention. To improve clarity, we have amended the text in the relevant sections (see lines 704-706, and legend of Figure 7 Supplement 1)

Minor issues:The terms in the title, "A single-cell transcriptomic atlas..." are used. What is meant by "atlas"? A searchable database or website is not provided.

Please refer to our response in the major comments section, where we explain that we have made our dataset accessible through a searchable web interface (https://scrna-analysiszebrafish.shinyapps.io/scatlas/) which is now referenced in the Data Availability Statement.

This reviewer considers that it is offensive to use terminology such as "poorly characterized" in reference to others' work.

Thank you for pointing this out. We understand the concern and have revised the wording to ensure it remains respectful and neutral when referring to previous work. The changes are reflected in lines 20 and 49.

The introduction of this manuscript should consider restructuring and editing. Example: Lines 51-57 introduce the importance of immune cells in zebrafish regeneration studies. However, this study does not investigate such processes. Additionally, the authors focus on the concept of immune heterogeneity in the brain throughout the text however, these studies have been conducted previously by others (Silva et al., 2021) at single-cell level.The novelty of this manuscript is the identification of "dendritic-like cells" and yet the introduction and text are limited to 68-71 lines. The introduction would benefit by introducing this cell type "dendritic-like cells" and differences between vertebrates.

Thank you for these valuable comments. In response, we have revised the introduction to better align with the focus of the study (see edited text in page 2). We now emphasize that, while macrophages have been extensively studied in zebrafish, dendritic cells remain much less well characterized in this model. Also, while we acknowledge that Silva et al. addressed aspects of immune heterogeneity in the zebrafish brain, their study primarily focused on mononuclear phagocytes. In contrast, our work provides a broader and more detailed characterization of the brain immune landscape, integrating transcriptomic data with multiple fluorescent reporter lines and hematopoietic mutants to strengthen cell identity assignments. Importantly, we note that Silva et al. classified DC-like cells within the microglial compartment, whereas our findings support that these cells represent a distinct population. While our data challenge this specific aspect of their conclusions, we believe both studies offer complementary insights that collectively advance our understanding of zebrafish brain immunity.

Though Figure 6 is a great conformation of scRNA sequencing, it seems redundant and should be supplemental data.

We respectfully disagree with the reviewer’s suggestion. We believe that presenting the data in Figure 6 as the main figure enhances its visibility and impact, particularly highlighting the distinction between microglia and DC-like cells, an aspect we consider highly valuable information for the zebrafish research community. This is especially important given that our conclusions challenge two previous independent reports, further underscoring the relevance of these findings to the field.